# Predicting Gas-Particle Partitioning Coefficients of Atmospheric Molecules with Machine Learning

Emma Lumiaro[1], Milica Todorović[1], Theo Kurten[2], Hanna Vehkamäki[3], and Patrick Rinke[1]

[1]Department of Applied Physics, Aalto University, P.O. Box 11100, 00076 Aalto, Espoo, Finland
[2]Department of Chemistry, Faculty of Science, PO Box 55, FI-00014 University of Helsinki, Finland
[3]Institute for Atmospheric and Earth System Research/Physics, Faculty of Science, PO Box 64, FI-00014 University of Helsinki, Finland

**Correspondence:** Patrick Rinke (patrick.rinke@aalto.fi)

**Abstract.** The formation, properties and lifetime of secondary organic aerosols in the atmosphere are largely determined by gas-particle partitioning coefficients of the participating organic vapours. Since these coefficients are often difficult to measure and to compute, we developed a machine learning model to predict them given molecular structure as input. Our data-driven approach is based on the dataset by Wang et al. (Atmos. Chem. Phys., 17, 7529 (2017)), who computed the partitioning coefficients and saturation vapour pressures of 3414 atmospheric oxidation products from the master chemical mechanism using the COSMOtherm program. We trained a kernel ridge regression (KRR) machine learning model on the saturation vapour pressure ($P_{sat}$), and on two equilibrium partitioning coefficients: between a water-insoluble organic matter phase and the gas phase ($K_{WIOM/G}$), and between an infinitely dilute solution with pure water and the gas phase ($K_{W/G}$). For the input representation of the atomic structure of each organic molecule to the machine, we tested different descriptors. We find that the many-body tensor representation (MBTR) works best for our application, but the topological fingerprint (TopFP) approach is almost as good, and computationally cheaper to evaluate. Our best machine learning model (KRR with a Gaussian kernel + MBTR) predicts $P_{sat}$ and $K_{WIOM/G}$ to within 0.3 logarithmic units and $K_{W/G}$ to within 0.4 logarithmic units of the original COSMOtherm calculations. This is equal or better than the typical accuracy of COSMOtherm predictions compared to experimental data (where available). We then applied our machine learning model to a dataset of 35,383 molecules that we generated based on a carbon 10 backbone functionalized with 0 to 6 carboxyl, carbonyl or hydroxyl groups to evaluate its performance for polyfunctional compounds with potentially low $P_{sat}$. The resulting saturation vapor pressure and partitioning coefficient distributions were physico-chemically reasonable, for example, in terms of the average effects of the addition of single functional groups. The volatility predictions for the most highly oxidized compounds were in qualitative agreement with experimentally inferred volatilities of, for example, alpha-pinene oxidation products with as-yet unknown structures, but similar elemental compositions.

# 1 Introduction

Aerosols in the atmosphere are fine solid or liquid particles (or droplets) suspended in air. They scatter and absorb solar radiation and form cloud droplets in the atmosphere, affect visibility and human health and are responsible for large uncertainties in the study of climate change (IPCC 2013). Most aerosol particles are secondary organic aerosols (SOAs) that are formed by oxidation of volatile organic compounds (VOCs), which are in turn emitted into the atmosphere for example from plants or traffic (Shrivastava et al., 2017). Some of the oxidation products have volatilities low enough to condense. The formation, growth and lifetime of SOAs is governed largely by the concentrations, saturation vapour pressures ($P_{sat}$) and equilibrium partitioning coefficients of the participating vapours. While real atmospheric aerosol particles are extremely complex mixtures of many different organic and inorganic compounds (Elm et al., 2020), partitioning of organic vapours is by necessity usually modelled in terms of a few representative parameters. These include the (liquid or solid) saturation vapour pressure, and various partitioning coefficients (K) in representative solvents such as water or octanol. The saturation vapor pressure is a pure-compound property, which essentially describes how efficiently a molecule interacts with other molecules of the same type. In contrast, partitioning coefficients depend on activity coefficients, which encompass the interaction of the compound with representative solvents. Typical partitioning coefficients in chemistry include ($K_{W/G}$) for the partitioning between the gas phase and pure water (i.e. an infinitely dilute solution of the compound), and ($K_{O/W}$) for the partitioning between octanol and water solutions [1]. For organic aerosols, the partitioning coefficient between the gas phase and a model water-insoluble organic matter phase (WIOM; $K_{WIOM/G}$) is more appropriate than ($K_{O/G}$).

Unfortunately, experimental measurements of these partitioning coefficients are challenging, especially for multifunctional low-volatility compounds most relevant to SOA formation. Little experimental data is thus available for the atmospherically most interesting organic vapour species. For relatively simple organic compounds, typically with up to three or four functional groups, efficient empirical parametrizations have been developed to predict their condensation-relevant properties, for example saturation vapor pressures. Such parameterizations include poly-parameter linear free-energy relationships (ppLFERs) (Goss and Schwarzenbach, 2001; Goss, 2004, 2006), the GROup contribution Method for Henry's law Estimate (GROMHE) (Raventos-Duran et al., 2010), and SPARC Performs Automated Reasoning in Chemistry (SPARC) (Hilal et al., 2008), SIM-POL (Pankow and Asher, 2008), EVAPORATION (Compernolle et al., 2011), and Nannoolal (Nannoolal et al., 2008). Many of these parameterisations are available in a user-friendly format on the UManSysProp website (Topping et al., 2016). However, due to the limitations in the available experimental datasets on which they are based, the accuracy of such approaches typically degrades significantly once the compound contains more than three or four functional groups (Valorso et al., 2011).

Approaches based on quantum chemistry such as COSMO-RS (COnductor-like Screening MOdel for Real Solvents, Klamt and Eckert (2000, 2003); Eckert and Klamt (2002)), implemented for example in the COSMOtherm program, can calculate (liquid or subcooled liquid) saturation vapour pressures and partitioning coefficients also for complex polyfunctional compounds, albeit only with order-of-magnitude accuracy at best. While the maximum deviation for the saturation vapor pressure predicted for the 310 compounds included in the original COSMOtherm parametrization dataset is only a factor of 3.7 (Eckert

---

[1]The gas-octanol partitioning coefficient ($K_{O/G}$) can then to good approximation be obtained from these by division.

and Klamt, 2002), the error margins increase rapidly especially with the number of intramolecular hydrogen bonds. In a very recent study, Hyttinen *et al.* estimated that the COSMOtherm prediction uncertainty for the saturation vapor pressure and the partitioning coefficient increases by a factor of 5 for each intra-molecular hydrogen bond (Hyttinen et al., 2021). However, for many applications even this level of accuracy is extremely useful. For example, in the context of new-particle formation (often called nucleation) it is beneficial to know, if the saturation vapour pressure of an organic compound is lower than about $10^{-12}$ kPa, because then it could condense irreversibly onto preexisting nanometer-sized cluster (Bianchi et al., 2019). An even lower $P_{\mathrm{sat}}$ would be required for the vapour to form completely new particles. This illustrates the challenge in performing experiments on SOA-relevant species: a compound with a saturation vapour pressure of e.g. $10^{-8}$ kPa at room temperature would be considered non-volatile in terms of most available measurement methods – yet its volatility is far too high to allow nucleation in the atmosphere. For a review of experimental saturation vapor pressure measurement techniques relevant to atmospheric science we refer to Bilde et al. (2015).

COSMO-RS/COSMOtherm calculations are based on density functional theory (DFT). In the context of quantum chemistry they are therefore considered computationally tractable compared to high-level methods such as coupled cluster theory. Nevertheless, the application of COSMO-RS to complex polyfunctional organic molecules still entails a significant computational effort, especially due to the conformational complexity of these species that need to be taken into account appropriately. Overall, there could be up to $10^4 - 10^7$ different organic compounds in the atmosphere (not even counting most oxidation intermediates), which makes the computation of saturation vapour pressures and partitioning coefficients a daunting task (Shrivastava et al., 2019; Ye et al., 2016).

Here, we take a different approach compared to previous parametrization studies, and consider a data-science perspective (Himanen et al., 2019). Instead of assuming chemical or physical relations, we let the data speak for itself. We develop and train a machine learning model to extract patterns from available data and predict saturation vapour pressures as well as partitioning coefficients.

Machine learning has only recently spread into atmospheric science (Cervone et al., 2008; Toms et al., 2018; Barnes et al., 2019; Nourani et al., 2019; Huntingford et al., 2019; Masuda et al., 2019). Prominent applications include the identification of forced climate patterns (Barnes et al., 2019), precipitation prediction (Nourani et al., 2019), climate analysis (Huntingford et al., 2019), pattern discovery (Toms et al., 2018), risk assessment of atmospheric emissions (Cervone et al., 2008), and the estimation of cloud optical thicknesses (Masuda et al., 2019). In molecular and materials science, machine learning is more established and now frequently complements theoretical or experimental methods (Müller et al., 2016; Ma et al., 2015; Shandiz and Gauvin, 2016; Gómez-Bombarelli et al., 2016; Bartók et al., 2017; Rupp et al., 2018; Goldsmith et al., 2018; Meyer et al., 2018; Zunger, 2018; Gu et al., 2019; Schmidt et al., 2019; Jensen et al.; Coley et al.). Here we build on our experience in atomistic, molecular machine learning (Ghosh et al., 2019; Todorović et al., 2019; Stuke et al., 2019; Himanen et al., 2020; Fang et al., 2021) to train a regression model that maps molecular structures onto saturation vapour pressures and partitioning coefficients. Once trained, the machine learning model can make saturation vapour pressure and partitioning predictions at COSMOtherm accuracy for hundreds of thousands of new molecules at no further computational cost. When

experimental training data becomes available, the machine learning model could easily be extended to encompass predictions for experimental pressures and coefficients.

Due to the above-mentioned lack of comprehensive experimental databases for saturation vapour pressures or gas-liquid partioning coefficient of polyfunctional atmospherically relevant molecules, our machine-learning model is based on the computational data by Wang et al. (2017). They computed the partitioning coefficients and saturation vapour pressures for 3414 atmospheric secondary oxidation products, obtained from the Master Chemical Mechanism (Jenkin et al., 1997; Saunders et al., 2003), using a combination of quantum chemistry and statistical thermodynamics as implemented in the COSMOtherm

approach (Klamt and Eckert, 2000). The parent VOCs for the MCM dataset include most of the atmospherically relevant small alkanes (methane, ethane, propane etc), alcohols, aldehydes, alkenes, ketones and aromatics, as well as chloro- and hydrochlorocabons, esters, ethers, and a few representative larger VOCs such as three monoterpenes and one sesquiterpene. Some inorganics are also included. For technical details on the COSMOtherm calculations performed by Wang et al., we refer to the COSMOtherm documentation (Klamt and Eckert, 2000), (Klamt, 2011), and a recent study by (Hyttinen et al., 2020), where the

conventions, definitions and notations used in COSMOtherm are connected to those more commonly employed in atmospheric physical chemistry. We note especially that the saturation vapor pressures computed by COSMOtherm correspond to the subcooled liquid state, and that the partitioning coefficients correspond to partitioning between two flat bulk surfaces in contact with each other. Actual partitioning between, e.g., aerosol particles and the gas phase will depend on further thermodynamic and kinetic parameters, which are not included here.

We transform the molecular structures in Wang's dataset into atomistic descriptors more suitable for machine learning than the atomic coordinates or the commonly used simplified molecular-input line-entry system (SMILES) strings. Optimal descriptor choices have been the subject of increased research in recent years (Langer et al., 2020; Rossi and Cumby, 2020; Himanen et al., 2020). We here test several descriptor choices: the many body tensor representation (Huo and Rupp, 2017), the Coulomb matrix (Rupp et al., 2012), the Molecular ACCess System (MACCS) structural key (Durant et al., 2002), a topological

fingerprint developed by RDkit (Landrum et al., 2006) based on the daylight fingerprint (James et al., 1995) and the Morgan fingerprint (Morgan, 1965).

Our work addresses the following objectives: 1) With view to future machine learning applications in atmospheric science, we assess the predictive capability of different structural descriptors for machine learning the chosen target properties. 2) We quantify the predictive power of our machine learning model for Wang's dataset to ascertain, if the dataset size is sufficient for

accurate machine learning predictions. 3) We then apply our validated machine learning model to a new molecular dataset to gain chemical insight into SOA condensation processes.

The paper is organized as follows. We describe our machine learning methodology in section 2, then present the machine learning results in section 3. Section 4 demonstrates how we employed the trained model for fast prediction of molecular properties. We discuss our findings and present a summary in section 5.

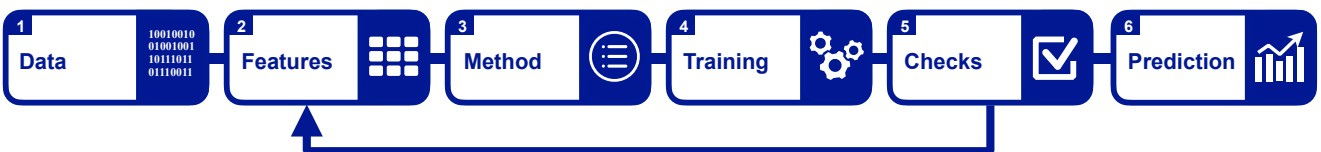

**Figure 1.** Schematic of our machine learning workflow: The raw input data is converted into molecular representations (referred to as features in this figure). We then set up and train a machine learning method. After evaluating its performance in step 5, we may adjust the features. Once the machine learning model is calibrated and trained, we make predictions on new data.

## 2 Methods

Our machine learning approach has six components as illustrated in Fig. 1. We start off with the raw data, which we present and analyse in section 2.1. The raw data is then transformed into a suitable representation[2] for machine learning (step 2). We introduce five different representations in section 2.2, which we test in our machine learning model (cf section 3). Next we choose our machine learning method. Here we use kernel ridge regression (KRR), which is introduced in section 2.3. After the machine learning model is trained in step 4, we analyse its learning success in step 5. The results of this process are shown in section 3. In this step we also make adjustments to the representation and the parameters of the model to improve the learning (see arrow from Checks to Features in Fig. 1). Finally, we use the best machine learning model to make predictions as shown in section 4.

### 2.1 Dataset

In this work we are interested in the the equilibrium partitioning coefficients of a molecule between a water-insoluble organic matter (WIOM) phase and its gas phase ($K_{\text{WIOM/G}}$) as well as between the gas phase and an infinitely diluted water solution. These coefficients are defined as

$$K_{\text{WIOM/G}} = \frac{C_{\text{WIOM}}}{C_{\text{G}}} \tag{1}$$

$$K_{\text{W/G}} = \frac{C_{\text{W}}}{C_{\text{G}}}, \tag{2}$$

where $C_{\text{WIOM}}$, $C_{\text{W}}$, and $C_{\text{G}}$ are the equilibrium concentrations of the molecule in the WIOM, water, and gas phase, respectively, at the limit of infinite dilution. In the framework of COSMOtherm calculations, gas-liquid partitioning coefficients can be converted into saturation vapor pressures, or vice versa, using the activity coefficients $\gamma_{\text{W}}$ or $\gamma_{\text{WIOM}}$ in the corresponding liquid (which can also be computed by COSMOtherm). Specifically, if for example $K_{\text{W/G}}$ is expressed in units of $m^3 g^{-1}$, then $K_{\text{W}} = \frac{RT}{M \gamma_{\text{W}} P_{\text{sat}}}$ , where $R$ is the gas constant, $T$ the temperature, $M$ the molar mass of the compound and $K_{\text{W}}$ and $\gamma_{\text{W}}$ the partitioning and activity coefficients in water (Arp and Goss, 2009). This illustrates that unlike the saturation vapor pressure $P_{\text{sat}}$, which is a pure-compound property, the partitioning coefficient also depends on the activity of the molecule in

---

[2]We use the words representation and features interchangeably.

the chosen liquid solvent, in this case water. We caution, however, that many different conventions exist e.g. for the dimensions of the partitioning coefficients, as well as the reference states for activity coefficients – the relation given above applies only to the particular conventions used by COSMOtherm. We refer to Hyttinen et al. (2020) for a discussion on the connection between different conventions and the notation used by COSMOtherm, and those commonly employed in atmospheric physical chemistry.

Wang et al. (2017) used the conductor-like screening model for real solvents (COSMO-RS) theory (Klamt and Eckert, 2000; Klamt, 2011) implemented in COSMOtherm to calculate the two partitioning coefficients $K_{WIOM/G}$[3] and $K_{W/G}$ for 3414 molecules. These molecules were generated from 143 parent volatile organic compounds with the Master Chemical Mechanism (MCM) (Jenkin et al., 1997; Saunders et al., 2003) through photolysis and reactions with ozone, hydroxide radicals and nitrate radicals.

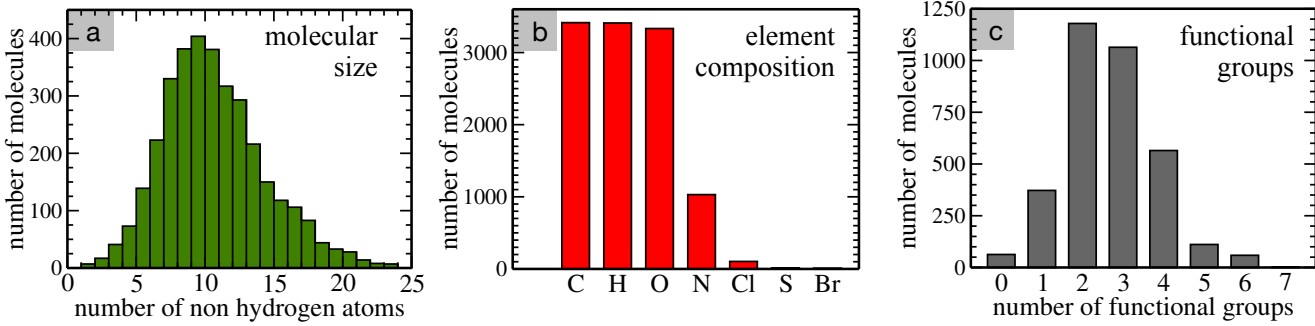

**Figure 2.** Dataset statistics: Panel a) shows the size distribution (in terms of the number of non-hydrogen atoms) of all 3414 molecules in the dataset. Panel b) illustrates how many molecules contain each of the chemical species and panel c) depicts the functional group distribution.

Here, we analyse the composition of the publicly available dataset by Wang *et al.* in preparation for machine learning. Figure 2 illustrates key dataset statistics. Panel a) shows the size distribution of molecules as measured in the number of non-hydrogen atoms. The 3414 non-radical species obtained from MGM range in size from 4 to 48 atoms, which translates into 2 to 24 non-hydrogen atoms per molecule. The distribution peaks at 10 non-hydrogen atoms and is skewed towards larger molecules. Panel b) illustrates how many molecules contain at least one atom of the indicated element. All molecules contain carbon (100% C), 3410 contain hydrogen (H; 99.88%) and 3333 also oxygen (O; 97.63%). Nitrogen (N) is the next most abundant element (30.17%) followed by chlorine (Cl; 3.05%), sulphur (S; 0.44%) and bromine (Br; 0.32%). Lastly, panel c) presents the distribution of functional groups. It peaks at 2 (34%) to 3 (31%) functional groups per molecule, with relatively few molecules having 0 (2%), 5 (3%) or 6 (2%) functional groups. The percentages for 1 and 4 functional groups are 11% and 17%, respectively.

---

[3]As a model WIOM phase Wang *et al.* used a compound originally suggested by Kalberer et al. (2004) as a representative secondary organic aerosol constituent. The IUPAC name for the compound in question, with elemental composition $C_{14}H_{16}O_5$, is 1-(5-(3,5-dimethylphenyl)dihydro-[1,3]dioxolo[4,5-d][1,3]dioxol-2-yl)ethan-1-one.

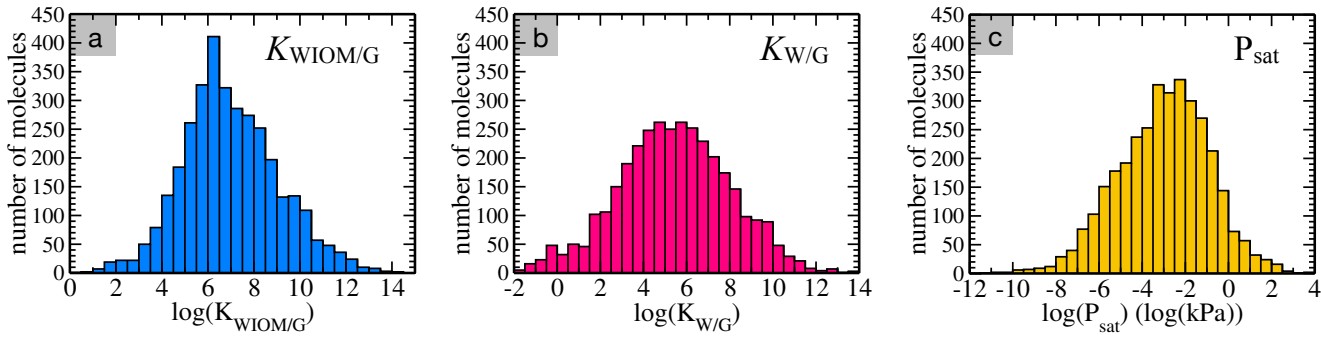

**Figure 3.** Dataset statistics: distributions of equilibrium partitioning coefficients a) $K_{WIOM/G}$, b) $K_{W/G}$ and c) the saturation vapour pressure $P_{sat}$ for all 3414 molecules in the dataset.

Figure 3 shows the distribution of the target properties $K_{WIOM/G}$, $K_{W/G}$ and $P_{sat}$ in Wang's dataset on a logarithmic scale. The equilibrium partitioning coefficient $K_{WIOM/G}$ distribution is skewed slightly towards larger coefficients, in contrast to the saturation vapour pressure $P_{sat}$ distribution that exhibits an asymmetry towards molecules with lower pressures. All three target properties cover approximately 15 logarithmic units and are approximately Gaussian distributed. Such peaked distributions are often not ideal for machine learning since they over-represent molecules near the peak of the distribution and under-represent molecules at their edges. The data peak does supply enough similarity to ensure good quality learning, but properties of the under-represented molecular types might be harder to learn.

We found 11 duplicate entries in Wang's dataset. These are documented in Section A in Tab. A1. The entries have the same SMILES strings and chemical formula, but differ in their Master Chemical Mechanism ID. Also the three target properties differ slightly. These duplicates did not affect the learning quality, so we did not remove them from the dataset.

Wang's dataset of 3414 molecules is relatively small for machine learning, which often requires hundreds of thousands to millions of training samples (Pyzer-Knapp et al., 2015; Smith et al., 2017; Stuke et al., 2019; Ghosh et al., 2019). A slightly larger set of Henry's law constants, which are related to $K_{W/G}$, were reported by Sander (2015) for 4632 organic species. Sander's database is a collection of 17350 Henry's law constant values collected from 689 references and therefore not as internally consistent as Wang's dataset. For example, the Sander dataset contains several molecules with multiple entries for the same property, sometimes spanning many orders of magnitude. We are not aware of a larger dataset that reports partitioning coefficients. For this reason, we rely exclusively on Wang's dataset and show that we can develop machine learning methods that are just as accurate as the underlying calculations and thus suitable for predictions.

## 2.2 Representations

The molecular representation for machine learning should fulfil certain requirements. It should be invariant with respect to translation and rotation of the molecule and permutations of atomic indices. Furthermore, it should be continuous, unique, compact and efficient to compute (Faber et al., 2015; Huo and Rupp, 2017; Langer et al., 2020; Himanen et al., 2020).

In this work we employ two classes of representations for the molecular structure, also known as descriptors: *physical* and *cheminformatics* descriptors. *Physical descriptors* encode physical distances and angles between atoms in the material or molecule. Meanwhile, decades of research in *cheminformatics* have produced topological descriptors that encode the qualitative aspects of molecules in a compact representation. These descriptors are typically bitvectors, in which molecular features are encoded (hashed) into binary fingerprints, which are joined into long binary vectors. In this work, we use two physical descriptors, the Coulomb Matrix and the many-body tensor, and three cheminformatics descriptors: the MACCS structural key, the topological fingerprint and the Morgan fingerprint.

In Wang's dataset the molecular structure is encoded in SMILES (Simplified Molecular Input Line Entry Specification) strings. We convert these SMILES strings into structural descriptors using Open Babel (O'Boyle et al., 2011) and the DScribe library (Himanen et al., 2020) or into cheminformatics descriptors using RDkit (Landrum et al., 2006).

### 2.2.1 Coulomb Matrix

The Coulomb matrix (CM) descriptor is inspired by an electrostatic representation of a molecule (Rupp et al., 2012). It encodes the cartesian coordinates of a molecule in a simple matrix of the form

$$C_{ij} = \begin{cases} 0.5 Z_i^{2.4} & \text{if } i = j \\ \frac{Z_i Z_j}{\|\mathbf{R}_i - \mathbf{R}_j\|} & \text{if } i \neq j \end{cases} \tag{3}$$

where $\mathbf{R}_i$ is the coordinate of atom $i$ with atomic charge $Z_i$. The diagonal provides element-specific information. The coefficient and the exponent have been fitted to the total energies of isolated atoms (Rupp et al., 2012). Off-diagonal elements encode inverse distances between the atoms of the molecule by means of a Coulomb-repulsion-like term.

The dimension of the Coulomb matrix is chosen to fit the largest molecule in the data set, i.e. it corresponds to the number of atoms of the largest molecule. The "empty" rows of Coulomb matrices for smaller molecules are padded with zeroes. Invariance with respect to the permutation of atoms in the molecule is enforced by simultaneously sorting rows and columns of each Coulomb matrix in descending order according to their $\ell^2$-norms. An example of a Coulomb matrix for 2-hydroxy-2-methylpropanoic acid is shown in Fig. 4b.

The CM is easily understandable, simple and relatively small as a descriptor. However, it performs best with Laplacian kernels in the machine-learning model (see Section 2.3), while other descriptors work better with the more standard choice of a Gaussian kernel.

### 2.2.2 Many-body tensor representation

The many-body tensor representation (MBTR) follows the Coulomb matrix philosophy of encoding the internal coordinates of a molecule. We will here describe the MBTR only qualitatively. Detailed equations can be found in the original publication (Huo and Rupp, 2017), our previous work (Himanen et al., 2020; Stuke et al., 2020) or Appendix B.

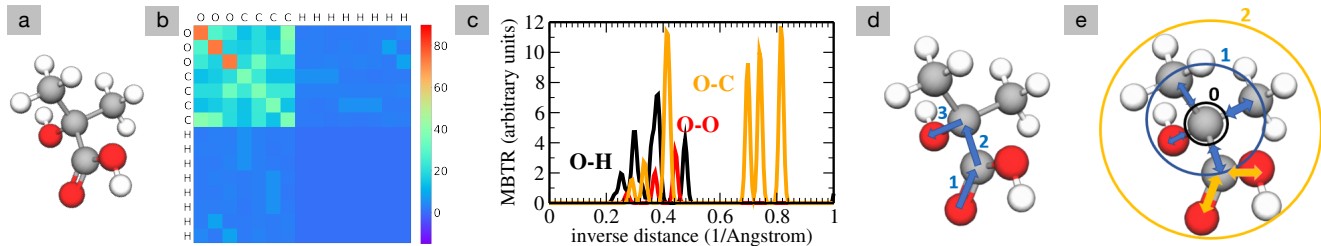

**Figure 4.** Pictorial overview over descriptors used in this work: a) ball and stick model of 2-hydroxy-2-methylpropanoic acid, b) corresponding Coulomb matrix (CM), c) the O-H, O-O and O-C inverse distance entries of the many-body tensor representation (MBTR), d) topological fingerprint (TopFP) depiction of a path with length three, and e) Morgan circular fingerprint with radius 0 (black), radius 1 (blue) and radius 2 (orange).

Unlike the Coulomb matrix, the many-body tensor is continuous and it distinguishes between different types of internal coordinates. At many-body level 1, the MBTR records the presence of all atomic species in a molecule by placing a Gaussian at the atomic number on an axis from 1 to the number of elements in the periodic table. The weight of the Gaussian is equal to the number of times the species is present in the molecule. At many-body level 2, inverse distances between every pair of atoms (bonded and non-bonded) are recorded in the same fashion. Many-body level 3 adds angular information between any triple of atoms. Higher levels (e.g. dihedral angles) would in principle be straightforward to add, but are not implemented in the current MBTR versions (Huo and Rupp, 2017; Himanen et al., 2020). Figure 4c shows selected MBTR elements for 2-hydroxy-2-methylpropanoic acid.

The MBTR is a continuous descriptor, which is advantageous for machine learning. However, MBTR is by far the largest descriptor out of the five we tested, and this can pose restrictions on memory and computational cost. Furthermore, the MBTR is more difficult to interpret than the CM.

### 2.2.3 MACCS Structural Key

The Molecular ACCess System (MACCS) structural key is a dictionary-based descriptor (Durant et al., 2002). It is represented as a bitvector of Boolean values that encode answers to a set of predefined questions. The MACCS structural key we used is a 166 bit long set of answers to 166 questions such as "Is there an S-S bond" or "Does it contain Iodine?" (Landrum et al., 2006; James et al., 1995).

MACCS is the smallest out of the five descriptors and extremely fast to use. Its accuracy critically depends on how well the 166 questions encapsulate the chemical detail of the molecules. Is it likely to reach a moderate accuracy with low computational cost and memory usage and could be beneficial for fast testing of a machine learning model.

### 2.2.4 Topological Fingerprint

The topological fingerprint (TopFP) is RDKit's original fingerprint (Landrum et al., 2006) inspired by the Daylight fingerprint (James et al., 1995). TopFP first extracts all topological paths of a certain lengths. The paths start from one atom in a molecule and travel along bonds until $k$ bond lengths have been traversed as illustrated in Fig. 4d. The path depicted in the figure would be OCCO. The list of patterns produced is exhaustive: Every pattern in the molecule, up to the pathlength limit, is generated. Each pattern then serves as a seed to a pseudo-random number generator (it is "hashed"), the output of which is a set of bits (typically 4 or 5 bits per pattern). The set of bits is added (with a logical OR) to the fingerprint. The length of the bitvector, maximum and minimum possible path lengths $k_{max}$ and $k_{min}$ and the length of one hash can be optimized.

Topology is an informative molecular feature. We therefore expect TopFP to balance good accuracy with reasonable computational cost. However, this binary fingerprint is difficult to visualize and analyse for chemical insight.

### 2.2.5 Morgan Fingerprint

The Morgan fingerprint is also a bit-vector constructed by hashing the molecular structure. In contrast to the Topological fingerprint, the Morgan fingerprint is hashed along circular or spherical paths around the central atom as illustrated in Figure 4e. Each substructure for a hash is constructed by first numbering the atoms in a molecule with unique integers by applying the Morgan algorithm. Each uniquely numbered atom then becomes a cluster center, around which we iteratively increase a spherical radius to include the neighbouring bonded atoms (Rogers and Hahn, 2010). Each radius increment extends the neighbour list by another molecular bond. The "circular" substructures found by the algorithm described above, excluding duplicates, are then hashed into a fingerprint (James et al., 1995; Landrum et al., 2006). The length of the fingerprint and the maximum radius can be optimized.

The Morgan fingerprint is quite similar to the TopFP in size and type of information encoded, so we expect similar performance. It also does not lend itself to easy chemical interpretation.

## 2.3 Machine Learning Method

### 2.3.1 Kernel Ridge Regression

In this work, we apply the kernel ridge regression (KRR) machine learning method. KRR is an example of supervised learning, in which the machine learning model is trained on pairs of input ($x$) and target ($f$) data. The trained model then predicts target values for previously unseen inputs. In this work, the input $x$ are the molecular descriptors CM and MBTR as well as the MACCS, TopFP and Morgan fingerprints. The targets are scalar values for the equilibrium partitioning coefficients and saturation vapour pressures.

KRR is based on Ridge Regression, in which a penalty for overfitting is added to an ordinary least squares fit (Friedman et al., 2001). In KRR, unlike Ridge regression, a nonlinear kernel is applied. This maps the molecular structure to our target properties in a high dimensional space (Stuke et al., 2019; Rupp, 2015).

The target values $f$ are a linear expansion in kernel elements

$$f(x) = \sum_{i=1}^{n} \alpha_i k(x_i, x),$$ (4)

where the sum runs over all training molecules. In this work, we use two different kernels, the Gaussian kernel

$$k_G(x, x') = e^{-\gamma \|x - x'\|_2^2}$$ (5)

and the Laplacian kernel

$$k_L(x, x') = e^{-\gamma \|x - x'\|_1}.$$ (6)

The kernel width $\gamma$ is a hyperparameter of the KRR model.

The regression coefficients $\alpha_i$ can be solved by minimizing the error

$$\min_{\alpha} \sum_{i=1}^{n} (f(x_i) - y_i)^2 + \lambda \boldsymbol{\alpha}^T \mathbf{K} \boldsymbol{\alpha},$$ (7)

where $y_i$ are reference target values for molecules in the training data. The second term is the regularization term, whose size is controlled by the hyperparameter $\lambda$. $\mathbf{K}$ is the kernel matrix of training inputs $k(x_i, x_j)$.

This minimization problem can be solved analytically for the expansion coefficients $\alpha_i$

$$\boldsymbol{\alpha} = (\mathbf{K} - \lambda \mathbf{I})^{-1} \mathbf{y}$$ (8)

The hyperparameters $\gamma$ and $\lambda$ need to be optimised separately.

We implemented KRR in Python using *scikit-learn* (Pedregosa et al., 2011). Our implementation has been described in Refs. (Stuke et al., 2019, 2020).

### 2.3.2 Computational Execution

Data used for supervised machine learning is typically divided into two sets, a large training set and a small test set. Both sets consists of input vectors and corresponding target properties. The KRR model is trained on the training set and its performance is quantified on the test set. At the outset, we separate a test set of 414 molecules. From the remaining molecules, we choose six different training sets of size 500, 1000, 1500, 2000, 2500 and 3000, so that a smaller training size is always a subset of the larger one. Training the model on a sequence of such training sets allows us to compute a *learning curve*, which facilitates the

assessment of learning success with increasing training data size. We quantify the accuracy of our KRR model by computing the mean absolute error (MAE) for the test set. To get statistically meaningful results, we repeat the training procedure 10 times. In each run, we shuffle the dataset before selecting the training and test sets so that the KRR model is trained and tested on different data each time. Each point on the learning curves is computed as the average over 10 results, and the spread serves as the standard deviation of the datapoint.

Model training proceeds by computing the KRR regression coefficients $\alpha_i$, obtained by minimizing equation 7. KRR hyperparameters $\gamma$ and $\lambda$ are typically optimized via grid search, and average optimal solutions are obtained by cross-validating the procedure. In cross-validation we split off a validation set from the training data before training the KRR model. KRR is then trained for all possible combinations of discretised hyperparameters (grid search) and evaluated on the validation set. This is done several times, so that the molecules in the validation set are changed each time. Then the hyperparameter combination with minimum average cross-validation error is chosen. Our implementation of cross-validated grid search is also based on Scikit-learn (Pedregosa et al., 2011). The optimized values for $\gamma$ and $\lambda$ are listed in Tab. B2.

**Table 1.** All the hyperparameters that were optimized.

|  | Hyperparameters | Optimized Values |
|---|---|---|
| KRR | width of the kernel $\gamma$, regularization parameter $\lambda$ | descriptor-dependent |
| MBTR | broadening parameters $\sigma_2, \sigma_3$; weighting parameters $w_2, w_3$ | 0.0075, 0.1; 1.2, 0.8 |
| TopFP | vector length; maximum path length $k_{max}$ ; bits per hash | 8192; 8; 16 |
| Morgan | vector length; radius; | 2048; 2 |

Table 1 summarises all the hyperparameters optimised in this study, those for KRR and the molecular descriptors, and their optimal values. In grid search, we varied both $\gamma$ and $\lambda$ by ten values between $10^{-1}$ and $10^{10}$. In addition, we used two different kernels, Laplacian and Gaussian. We compared the performance of the two kernels for the average of 5 runs for each training size and the most optimal kernel was chosen. In cases in which both kernels performed equally well, e.g., for the fingerprints, we chose the Gaussian kernel for its lower computational cost.

To compute the MBTR and CM descriptors we employed the *openbabel* software to convert the SMILES strings provided in the Wang *et al.* dataset into 3-dimensional molecular structures. We did not perform any conformer search. MBTR hyperparameters and TopFP hyperparameters were optimized by grid search for several training set sizes (MBTR for sizes 500, 1500 and 3000 and TopFP for size 1000 and 1500 ) and the average of two runs for each training size was taken. We did not extend the descriptor hyperparameter search to larger training set sizes, since we found that the hyperparameters were insensitive to the training set size. The MBTR weighting parameters were optimized in 8 steps between 0 (no weighting) and 1.4, and the broadening parameters in 6 steps between $10^{-1}$ and $10^{-6}$. The length of TopFP was varied between 1024 and 8192 (size can varied by $2^n$). The range for the maximum path length extended from 5 to 11 and the bits per hash were varied between 3 and 16.

## 3 Results

In Figure 5 we present the learning curves for our objectives $K_{\mathrm{WIOM/G}}$, $K_{\mathrm{W/G}}$ and $P_{\mathrm{sat}}$. Shown is the mean average error (MAE) as a function of the training set size for all three target properties and for all five molecular descriptors. As expected, the MAE decreases as the training size increases. For all target properties, the lowest errors are achieved with MBTR and the

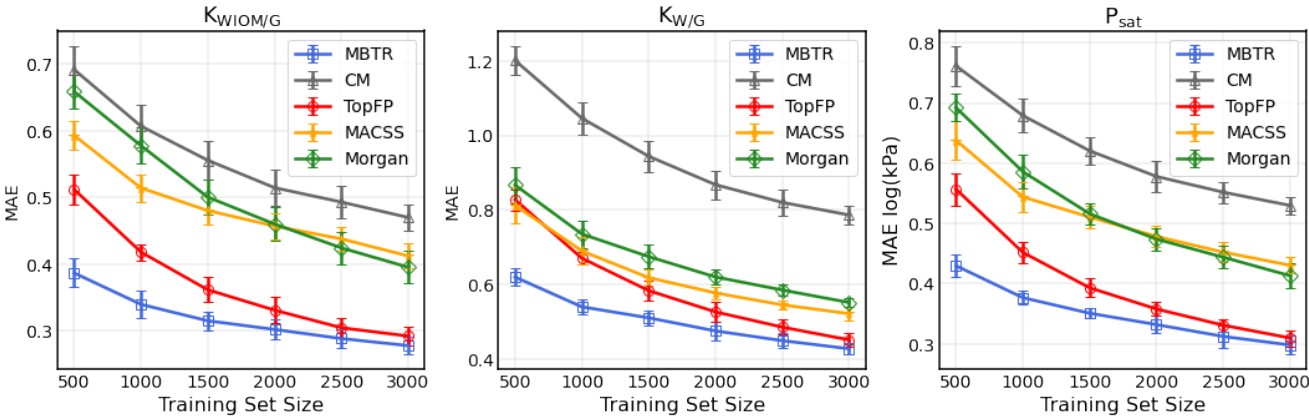

**Figure 5.** The learning curves for equilibrium partitioning coefficients $K_{\mathrm{WIOM/G}}$, $K_{\mathrm{W/G}}$ and saturation vapour pressure $P_{\mathrm{sat}}$ for predictions made with all five descriptors.

worst performing descriptor is CM. TopFP approaches the accuracy of MBTR as the training size increases and appears likely to outperform MBTR beyond the largest training size of 3000 molecules.

Table 2 summarises the average MAEs and their standard deviations for the best-trained KRR model (training size of 3000 with MBTR descriptor). The highest accuracy is obtained for partitioning coefficient $K_{\mathrm{WIOM/G}}$, with a mean average error of 0.278, i.e. only 1.9% of the entire $K_{\mathrm{WIOM/G}}$ range. The second best accuracy is obtained for saturation vapour pressure $P_{\mathrm{sat}}$ with an MAE of 0.298 (or 2.0% of the range of pressure values). The lowest accuracy is obtained for $K_{\mathrm{W/G}}$ with an MAE of 0.428. However, the range for partitioning coefficient $K_{\mathrm{W/G}}$ is also the largest, as seen in Figure 3, so this amounts to only 2.7% of the entire range of values. Our best machine learning MAEs are of the order of the COSMOtherm prediction accuracy, which lies at around a few tenths of log values (Stenzel et al., 2014; Schröder et al., 2016; van der Spoel et al., 2019).

Figure 6 shows the results for the best-performing descriptors MBTR and TopFP in more detail. The scatter plots illustrate how well the KRR predictions match the reference values. The match is further quantified by $R^2$ values. For all three target values, the predictions hug the diagonal quite closely and we observe only a few outliers that are further away from the diagonal. The predictions of partitioning coefficient $K_{\mathrm{WIOM/G}}$ are most accurate. This is expected because the MAE in Table 2 is lowest for this property. The largest scattered is observed for partitioning coefficient $K_{\mathrm{W/G}}$, which had the highest MAE in Table 2.

**Table 2.** The average mean average errors (MAE) and the standard deviations for all the descriptors and target properties (equilibrium partitioning coefficients $K_{\mathrm{WIOM/G}}$, $K_{\mathrm{W/G}}$ and saturation vapour pressure $P_{\mathrm{sat}}$) with the largest possible training size of 3000.

| | $K_{\mathrm{WIOM/G}}$ | | $K_{\mathrm{W/G}}$ | | $P_{\mathrm{sat}}$ | |
|---|---|---|---|---|---|---|
| Descriptor | MAE | $\Delta$ | MAE | $\Delta$ | MAE log(kPa) | $\Delta$log(kPa) |
| CM | 0.470 | $\pm\,0.020$ | 0.787 | $\pm\,0.028$ | 0.530 | $\pm\,0.016$ |
| MBTR | 0.278 | $\pm\,0.013$ | 0.427 | $\pm\,0.015$ | 0.298 | $\pm\,0.016$ |
| MACCS | 0.412 | $\pm\,0.020$ | 0.522 | $\pm\,0.020$ | 0.431 | $\pm\,0.014$ |
| Morgan | 0.396 | $\pm\,0.026$ | 0.552 | $\pm\,0.014$ | 0.413 | $\pm\,0.022$ |
| TopFP | 0.292 | $\pm\,0.014$ | 0.451 | $\pm\,0.021$ | 0.310 | $\pm\,0.014$ |

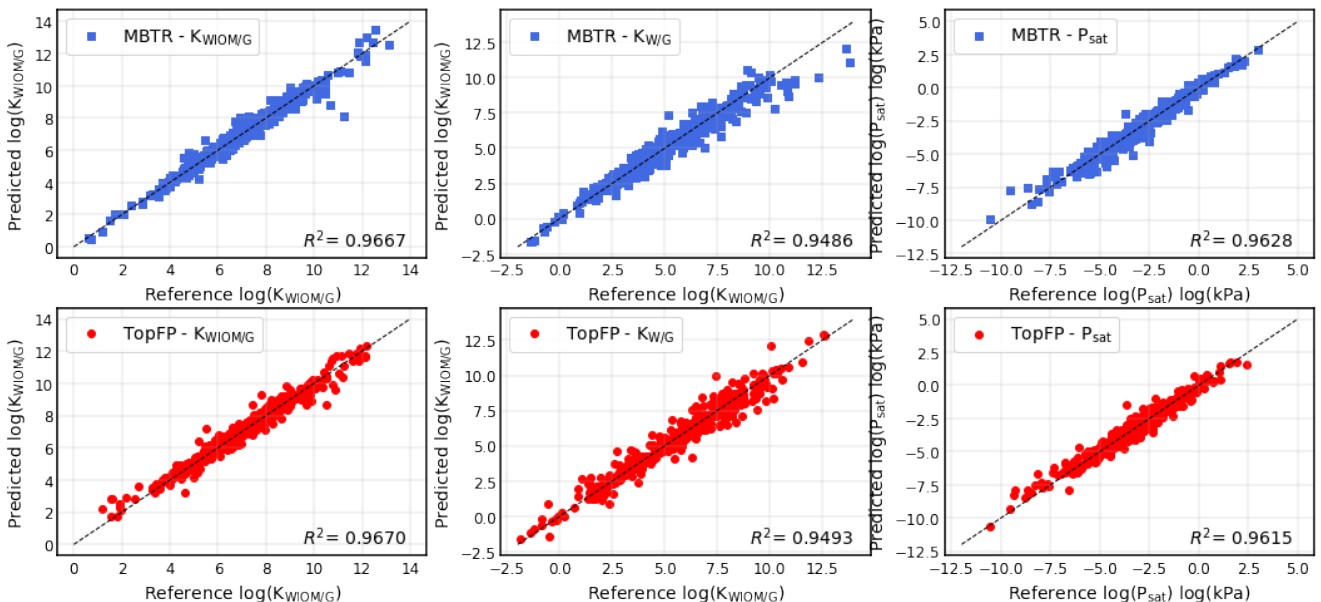

**Figure 6.** Scatter plots for predictions of the partitioning coefficients of a molecule between a water-insoluble organic matter and gas phase $K_{\mathrm{WIOM/G}}$, water and gas phase $K_{\mathrm{W/G}}$ and the saturation vapour pressure $P_{\mathrm{sat}}$ for the test set of 414 molecules using MBTR (top) and TopFP (bottom). The prediction with the lowest mean average error was chosen for each scatter plot.

## 4 Predictions

In the previous section we showed that our KRR model trained on the Wang *et al.* dataset produces low prediction errors for molecular partitioning coefficients and saturation vapour pressures and can now be employed as a fast predictor. When

shown further molecular structures, it can make instant predictions for the molecular properties of interest. We demonstrate this application potential on an example dataset generated to imitate organic molecules typically found in the atmosphere.

Atmospheric oxidation reaction mechanisms can be generally classified into two main types: fragmentation and function-
335 alization (Kroll et al., 2009; Seinfeld and Pandis, 2016). For SOA formation, functionalization is more relevant, as it leads to products with intact carbon backbones and added polar (and volatility-lowering) functional groups. Many of the most interesting molecules from a SOA-forming point of view, e.g. monoterpenes, have around 10 carbon atoms (Zhang et al., 2018). These compounds simultaneously have high enough emissions or concentrations to produce appreciable amounts of condensable products, while being large enough for those products to have low volatility.

We thus generated a dataset of molecules with a backbone of ten carbon (C10) atoms. For simplicity, we used a linear alkane chain. In analogy with Wang's dataset, we then decorated this backbone with 0 to 6 functional groups at different locations. We limited ourselves to the typical groups formed in "functionalizing" oxidation of VOC by both of the main day-time oxidants OH and $O_3$: carboxyl(-COOH), carbonyl (=O) and hydroxyl (-OH) (Seinfeld and Pandis, 2016). The (-COOH) group can only be added to the ends of the C10 molecule, while (=O) and (-OH) can be added to any carbon atom in the chain. We then generated
all possible combinations combinatorially and filtered out duplicates resulting from symmetric combinations of functional groups. In total we obtained 35,383 unique molecules. Example molecules are depicted in Figure 9. While the functional group composition of our C10 dataset is atmospherically relevant, the particular molecules are not. The purpose of this dataset is to perform a relatively simple sanity check on the machine learning predictions, on a set of compounds structurally different from those in the training dataset. We note that using e.g. more atmospherically relevant compounds such as alpha-pinene
oxidation products for this purpose might be counter-productive, since Wang *et al.*'s dataset used for training contains several such compounds.

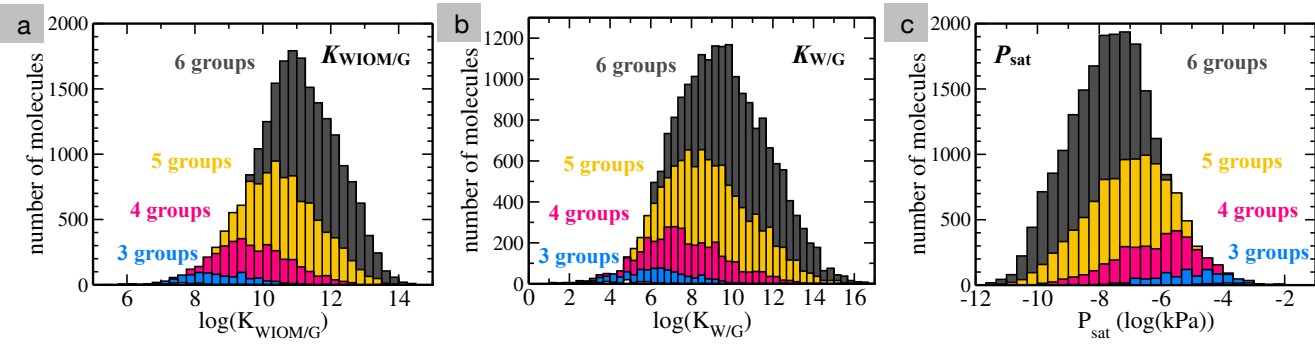

**Figure 7.** Histograms of C10 TopFP-KRR predictions for a) $K_{\mathrm{WIOM/G}}$,b ) $K_{W/G}$ and c) $P_{\mathrm{sat}}$. The histograms are divided into different numbers of functional groups. Molecules with 2 or fewer functional groups have been omitted from these histograms, because their total number is very low in the C10 dataset.

For each of the 35,383 molecules, we generated a SMILES string that serves as input for the TopFP fingerprint. We did not relax the geometry of the molecules with force fields or density-functional theory. We chose TopFP as descriptor, because

its accuracy is close to that of the best performing MBTR KRR model, but significantly cheaper to evaluate. TopFP is also invariant to conformer choices, since the fingerprint is the same for all conformers of a molecule. We then predicted $P_{\text{sat}}$, $K_{\text{WIOM/G}}$ and $K_{\text{W/G}}$ with the TopFP-KRR model.

Figures 7 and 8 show the predictions of our TopFP-KRR model for the C10 dataset. For comparison with Wang's dataset, we broke the histograms and analysis down by the number of functional groups. For a given number of functional groups, the partitioning coefficients for our C10 dataset are somewhat higher, and the saturation vapor pressures correspondingly somewhat lower, than in Wang's dataset. This follows from the fact that our C10 molecules are on average larger [4] than those contained in Wang's dataset (Figure 2). However, as seen from Figure 8, the averages of all three quantities (for a given number of functional groups) are not substantially different, illustrating the similarity of both datasets. A certain degree of similarity is required to ensure predictive power, since machine learning models do not extrapolate well to data that lies outside the training range.

The variation in the studied parameters is larger in Wang's dataset for molecules with 4 or less functional groups, but similar or smaller for molecules with 5 or 6 functional groups. This is likely the case, because Wang's dataset contains relatively few compounds with more than four functional groups. The variation in the studied parameters (for each number of functional groups) predicted for the C10 dataset is in line with the individual group contributions predicted based on fits to experimental data, for example, by the SIMPOL model (Pankow and Asher, 2008) for saturation vapor pressures. According to SIMPOL, a carboxylic acid group decreases the saturation vapor pressure at room temperature by almost a factor of 4000, while a ketone group reduces it by less than a factor of 9. Accordingly, if interactions between functional groups are ignored, a dicarboxylic acid, for example, should have a saturation vapor pressure more than 100 000 times lower than a diketone with the same carbon backbone. This is remarkably consistent with Figure 8, where the variation of saturation vapor pressures for compounds with two functional groups in our C10 dataset is slightly more than 5 orders of magnitude.

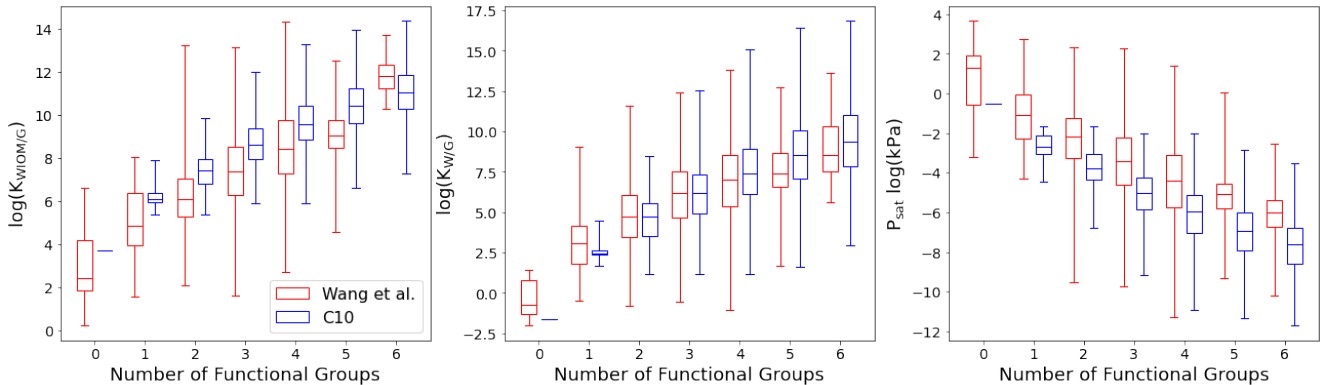

**Figure 8.** Box plot comparing C10 (in blue) with Wang's dataset (in red) for $K_{\text{WIOM/G}}$, $K_{\text{W/G}}$ and $P_{\text{sat}}$ for different numbers of functional groups. Shown are the minimum, maximum, median, first and third quartile.

---

[4] Our C10 molecules range in size from 10 to 18 non-hydrogen atoms since the largest of our molecules contains two carboxylic acid and four ketone and/or hydroxyl groups.

Figure 7 illustrates that the saturation vapour pressure $P_{\text{sat}}$ decreases with increasing number of functional groups as expected, whereas $K_{\text{WIOM/G}}$ and $K_{\text{W/G}}$ increase. This is consistent with Wang's dataset as shown in Fig. 8, where we compare averages between the two datasets. The magnitude of the decrease (increase) amounts to approximately 1 or 2 orders of magnitude per functional group and is, again, consistent with existing structure-activity relationships based on experimental data (e.g. Pankow and Asher (2008); Compernolle et al. (2011); Nannoolal et al. (2008)).

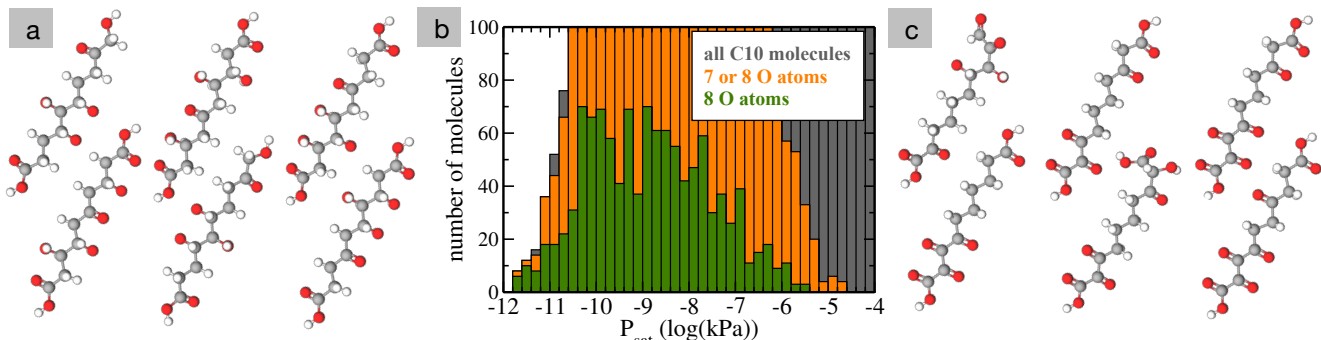

**Figure 9.** a) Atomic structure of the 6 molecules with the lowest predicted saturation vapour pressure $P_{\text{sat}}$; b) $P_{\text{sat}}$ histograms for molecules containing 7 or 8 O atoms (orange) or only 8 O atoms (green). For reference, the histogram of all molecules (grey) is also shown. c) Atomic structure of the 6 molecules with 7 and 8 O atoms and the highest saturation vapour pressure $P_{\text{sat}}$.

The region of low $P_{\text{sat}}$ is most relevant for atmospheric SOA formation. However, we caution that COSMOtherm predictions have not yet been properly validated against experiments for this pressure regime. As discussed above, we can hope for order-of-magnitude accuracy at best. Figure 9b shows histograms of only molecules with 7 or 8 oxygen atoms. These are compared to the full dataset. Since the "8 O atom set" is a subset of the "7 or 8 O atoms" set, which in turn is a subset of "all molecules" the lengths of the bars in a given bin reflect the percentages of molecules with 7 or 8 O atoms. We observe that below $10^{-10}$ kPa, almost all C10 molecules contain 7 or 8 O atoms, as there is little grey visible in that part of the histogram. In the context of atmospheric chemistry, the least-volatile fraction of our C10 dataset corresponds to LVOC ("low volatility organic compounds"), which are capable of condensing onto small aerosol particles, but not actually forming them. Our results are thus in qualitative agreement with recent experimental results by (Peräkylä et al., 2020), who concluded that the highly oxidized $C_{10}$ products of $\alpha$-pinene oxidation are mostly LVOC. However, we note that the compounds measured by Peräkylä et al. are likely to contain functional groups not included in our C10 dataset, as well as structural features such as branching and rings.

Figure 9a and Figure 9c show the molecular structures of the lowest-volatility compounds, as well as the highest-volatility compounds with 7 or 8 O atoms, respectively. The 6 shown highest volatility compounds inevitably contain at least one carboxylic acid group, as we have restricted the number of functional groups to six or less, and only the acid groups contain two oxygen atoms. Comparing the two sets, we see that the lowest-volatility compounds contain more hydroxyl groups, and less ketone groups, while the highest-volatility compounds with 7 or 8 oxygen atoms contain almost no hydroxyl groups. This is expected, since, e.g., a hydroxyl group lowers the saturation vapor pressure by over a factor of 100 at 298 K, while

the effect of a ketone group is, as previously noted, less than a factor of 9 according to the SIMPOL model (Pankow and Asher, 2008). However, even the lowest-volatility compounds (Figure 9a) contain a few ketone groups, such that the number of hydrogen-bond donor and acceptor groups are roughly similar. This result demonstrates that unlike the simplest group-contribution models such as SIMPOL, which would invariably predict that the lowest-volatility compounds in our C10 dataset

should be the tetrahydroxydicarboxylic acids, both the original COSMOtherm predictions, and the machine-learning model based on them, are capable of accounting for hydrogen-bonding interactions between functional groups. As we did not include conformational information of our C10 molecules in the machine-learning predictions, this is most likely due to structural similarities between the C10 compounds, and hydrogen-bonding molecules in the training dataset.

Lastly, we consider the issue of non-unique descriptors. Although the cheminformatics descriptors are fast to compute and

405 use, a duplicate check revealed that it is possible to obtain identical descriptors for different molecule structures, even for this relatively small pool of molecules. The MACCS fingerprint in particular produced over 500 duplicates (about 15% of the dataset) because its query list is not sufficiently descriptive of this molecule class. Some duplicates were also observed for TopFP ($<1.5\%$), whereas physical descriptors were both entirely unique, as expected. The original dataset itself contained 11 identical molecular structures labelled with different SMILES strings, as mentioned in Section 2.1. AI model checks revealed

that the number of duplicates in this study was small enough to have a negligible effect on predictions (apart from the MACCS key models) so we did not purge them.

## 5    Conclusions

In this study, we set out to evaluate the potential of the KRR machine learning method to map molecular structures to its atmospheric partitioning behaviour, and establish which molecular descriptor has the best predictive capability.

KRR is a relatively simple kernel-based machine-learning technique that is straightforward to implement and fast to train. Given model simplicity, the quality of learning depends strongly on information content of the molecular descriptor. More specifically, it hinges on how well each format encapsulates the structural features relevant to the atmospheric behaviour. The exhaustive approach of the MBTR descriptor to documenting molecular features has led to very good predictive accuracy in machine learning of molecular properties (Stuke et al., 2019; Langer et al., 2020; Rossi and Cumby, 2020; Himanen et al., 2020)

and this work is no exception. The lightweight CM descriptor does not perform nearly as well, but these two representations from physical sciences provide us with an upper and lower limit on predictive accuracy.

Descriptors from cheminformatics that were developed specifically for molecules have variable performance. Between them, the topological fingerprint leads to the best learning quality that approaches MBTR accuracy in the limit of larger training set sizes. This is a notable finding, not least because the relatively small TopFP data structures in comparison to MBTR reduce

the computational time and memory required for machine learning. MBTR encoding requires knowledge of the 3-dimensional molecular structure, which raises the issue of conformer search. It is unclear which molecular conformers are relevant for atmospheric condensation behaviour, and COSMOterm calculations on different conformers can produce values that are orders of magnitude apart. TopFP requires only connectivity information and can be built from SMILES strings, eliminating any

conformer considerations (albeit at the cost of possibly losing some information on e.g. intramolecular hydrogen bonds). All this makes TopFP the most promising descriptor for future machine learning studies in atmospheric science that we have identified in this work.

Our results show that KRR can be used to train a model to predict COSMOtherm saturation vapor pressures, with error margins smaller than those of the original COSMOtherm predictions. In the future, we will extend our training set to encompass especially atmospheric autoxidation products (Bianchi et al., 2019), which are not included in existing saturation vapour pressure datasets, and for which existing prediction methods are highly uncertain. We also intend to extend the machine learning model to predict a larger set of parameters computed by COSMOtherm, such as vaporization enthalpies, internal energies of phase transfer, and acivity coefficients in representative phases. While COSMOtherm predictions for complex molecules such as autoxidation products also have large uncertainties, a fast and efficient "COSMOtherm - level" KRR predictor would still be immensely useful, for example, for assessing whether a given compound is likely to have extremely low volatility or not. Experimental volatility data for such compounds is also gradually becoming available, either through indirect inference methods such as Peräkylä et al. (2020), or for example from thermal desorption measurements (Li et al., 2020). These can then be used to constrain and anchor the model, and ultimately yield also quantitatively reliable volatility predictions.

*Code and data availability.* The Wang dataset (Wang et al., 2017) and the novel C10 dataset of atmosperic molecules (Atmospheric C10 dataset, Zenodo , 2020) are freely available online. The KRR code employed in this study can be found on Gitlab (KRR for Atmospheric molecules, Gitlab , 2020).

**Appendix A: Data set duplicates**

**Table A1.** Duplicates found in Wang *et al.*'s dataset: listed are the index in the dataset, the ID in the Master Chemical Mechanism (MCM_ID), the corresponding SMILES string, the chemical formula and the three target properties ($K_{WIOM/G}$, $K_{W/G}$ and $P_{sat}$).

| # | index | MCM_ID | SMILES | Formula | $K_{WIOM/G}$ | $K_{W/G}$ | $P_{sat}$ |
|---|-------|--------|--------|---------|-------------|-----------|-----------|
| 1 | 83 | MACRNB | CC(C=O)(CON(=O)=O)O | C4H7NO5 | 5.51 | 3.42 | 4.33E-02 |
|   | 716 | MACROHNO3 | CC(C=O)(CON(=O)=O)O | C4H7NO5 | 5.59 | 3.5 | 3.44E-02 |
| 2 | 1943 | CHOMOHCO3H | CC(C=O)(C(=O)OO)O | C4H6O5 | 5.93 | 5.32 | 2.26E-02 |
|   | 84 | COHM2CO3H | CC(C=O)(C(=O)OO)O | C4H6O5 | 5.89 | 4.83 | 2.55E-02 |
| 3 | 439 | IEB1OOH | CC(C=O)(C(CO)O)OO | C5H10O5 | 7.22 | 8.02 | 1.64E-04 |
|   | 2624 | C57OOH | CC(C=O)(C(CO)O)OO | C5H10O5 | 7.24 | 7.47 | 1.90E-04 |
| 4 | 730 | MACRNBCO3H | CC(CON(=O)=O)(C(=O)O)O | C4H7NO6 | 8.06 | 6.85 | 2.86E-04 |
|   | 2469 | MACRNBCO2H | CC(CON(=O)=O)(C(=O)O)O | C4H7NO6 | 8.12 | 6.93 | 2.34E-04 |
| 5 | 817 | TDICLETH | C(=CCl)Cl | C2H2Cl2 | 2.54 | 0.41 | 4.15E+01 |
|   | 3141 | CDICLETH | C(=CCl)Cl | C2H2Cl2 | 2.54 | 0.41 | 4.15E+01 |
| 6 | 819 | CHOMOHPAN | CC(C=O)(C(=O)OON(=O)=O)O | C4H5NO7 | 5.49 | 2.31 | 1.43E-01 |
|   | 1221 | COHM2PAN | CC(C=O)(C(=O)OON(=O)=O)O | C4H5NO7 | 5.37 | 2.08 | 1.99E-01 |
| 7 | 900 | THEX2ENE | CC=CCCC | C6H12 | 2.4 | -0.92 | 1.95E+01 |
|   | 3372 | CHEX2ENE | CC=CCCC | C6H12 | 2.4 | -0.92 | 1.94E+01 |
| 8 | 1443 | CPENT2ENE | CC=CCC | C5H10 | 1.9 | -0.86 | 7.84E+01 |
|   | 3119 | TPENT2ENE | CC=CCC | C5H10 | 1.91 | -0.85 | 7.71E+01 |
| 9 | 1649 | CBUT2ENE | CC=CC | C4H8 | 1.5 | -0.84 | 2.56E+02 |
|   | 1665 | TBUT2ENE | CC=CC | C4H8 | 1.49 | -0.84 | 2.58E+02 |
| 10 | 2188 | CO2N3CHO | CC(=O)C(C=O)ON(=O)=O | C4H5NO5 | 5.16 | 2.62 | 1.22E-01 |
|   | 3040 | C4CONO3CO | CC(=O)C(C=O)ON(=O)=O | C4H5NO5 | 5.21 | 2.72 | 1.05E-01 |
| 11 | 2127 | C59OOH | CC(CO)(C(=O)CO)OO | C5H10O5 | 7.59 | 7.86 | 6.75E-05 |
|   | 2636 | IEC1OOH | CC(CO)(C(=O)CO)OO | C5H10O5 | 7.53 | 7.76 | 8.09E-05 |

## Appendix B: Many-body tensor representation

In this appendix we provide the mathematical structure of the MBTR as it is implemented in the DScribe library Himanen et al. (2020). The many-body levels in the MBTR are denoted $k$. For $k = 1, 2, 3$, geometry functions encode the different features: $g_1(Z_l) = Z_l$ (atomic number), $g_2(\boldsymbol{R}_l, \boldsymbol{R}_m) = |\boldsymbol{R}_l - \boldsymbol{R}_m|$ (distance) or $g_2(\boldsymbol{R}_l, \boldsymbol{R}_m) = \frac{1}{|\boldsymbol{R}_l - \boldsymbol{R}_m|}$ (inverse distance), and $g_3(\boldsymbol{R}_l, \boldsymbol{R}_m, \boldsymbol{R}_n) = \cos(\angle(\boldsymbol{R}_l - \boldsymbol{R}_m, \boldsymbol{R}_n - \boldsymbol{R}_m))$ (cosine of angle).

The scalar values returned by the geometry functions $g_k$ are Gaussian broadened into continuous representations $\mathcal{D}_k$:

$$\mathcal{D}_1^l(x) = \frac{1}{\sigma_1\sqrt{2\pi}}e^{-\frac{(x - g_1(Z_l))^2}{2\sigma_1^2}} \tag{B1}$$

$$\mathcal{D}_2^{l,m}(x) = \frac{1}{\sigma_2\sqrt{2\pi}}e^{-\frac{(x - g_2(\boldsymbol{R}_l, \boldsymbol{R}_m))^2}{2\sigma_2^2}} \tag{B2}$$

$$\mathcal{D}_3^{l,m,n}(x) = \frac{1}{\sigma_3\sqrt{2\pi}}e^{-\frac{(x - g_3(\boldsymbol{R}_l, \boldsymbol{R}_m), \boldsymbol{R}_n)^2}{2\sigma_3^2}} . \tag{B3}$$

The $\sigma_k$'s are the feature widths for the different $k$-levels and $x$ runs over a predefined range $[x_{\min}^k, x_{\max}^k]$ of possible values for the geometry functions $g_k$.

Finally, a weighted sum of distributions $\mathcal{D}_k$ is generated for each possible combination of chemical elements present in the dataset

$$\mathrm{MBTR}_1^{Z_1}(x) = \sum_l^{|Z_1|} w_1^l \mathcal{D}_1^l(x) \tag{B4}$$

$$\mathrm{MBTR}_2^{Z_1, Z_2}(x) = \sum_l^{|Z_1|}\sum_m^{|Z_2|} w_2^{l,m} \mathcal{D}_2^{l,m}(x) \tag{B5}$$

$$\mathrm{MBTR}_3^{Z_1, Z_2, Z_3}(x) = \sum_l^{|Z_1|}\sum_m^{|Z_2|}\sum_n^{|Z_3|} w_3^{l,m,n} \mathcal{D}_3^{l,m,n}(x). \tag{B6}$$

The sums for $l$, $m$, and $n$ run over all atoms with atomic numbers $Z_1$, $Z_2$ and $Z_3$. $w_k$ are weighting functions that balance the relative importance of different $k$-terms and/or limit the range of inter-atomic interactions. For $k = 1$, usually no weighting is used ($w_1^l = 1$). For $k = 2$ and $k = 3$ the following exponential decay functions are implemented in `DScribe`

$$w_2^{l,m} = e^{-s_k|\boldsymbol{R}_l - \boldsymbol{R}_m|} \tag{B7}$$

$$w_3^{l,m,n} = e^{-s_k(|\boldsymbol{R}_l - \boldsymbol{R}_m| + |\boldsymbol{R}_m - \boldsymbol{R}_n| + |\boldsymbol{R}_l - \boldsymbol{R}_n|)} \tag{B8}$$

**Table B1.** Values of the optimal KRR hyperparameter $\lambda$ obtained by cross-validation as a function of descriptor type and training set size. The procedure was repeated 10 times with re-shuffled data. Average values ($\bar{\lambda}$) were used in further KRR models. We also report the statistical standard deviation $\Delta\lambda$.

| Descriptor | Training Set Size | $K_{WIOM/G}$ | | $K_{W/G}$ | | $P_{sat}$ | |
|---|---|---|---|---|---|---|---|
| | | $\bar{\lambda}$ | $\Delta\lambda$ | $\bar{\lambda}$ | $\Delta\lambda$ | $\bar{\lambda}$ | $\Delta\lambda$ |
| CM | 500 | 0.00E+00 | 4.50E-03 | 9.10E-03 | 2.85E-03 | 1.00E-02 | 0.00E+00 |
| | 1000 | 8.20E-03 | 3.79E-03 | 5.50E-03 | 4.74E-03 | 1.00E-02 | 0.00E+00 |
| | 1500 | 5.50E-03 | 4.74E-03 | 3.70E-03 | 4.35E-03 | 5.50E-03 | 4.74E-03 |
| | 2000 | 3.70E-03 | 4.35E-03 | 1.81E-03 | 2.89E-03 | 5.50E-03 | 4.74E-03 |
| | 2500 | 2.80E-03 | 3.79E-03 | 1.00E-03 | 0.00E+00 | 1.90E-03 | 2.85E-03 |
| | 3000 | 1.90E-03 | 2.85E-03 | 1.00E-03 | 0.00E+00 | 1.90E-03 | 2.85E-03 |
| MBTR | 500 | 5.30E-05 | 4.97E-05 | 7.30E-05 | 4.35E-05 | 1.44E-04 | 3.04E-04 |
| | 1000 | 8.02E-05 | 4.17E-05 | 1.00E-04 | 0.00E+00 | 1.81E-04 | 2.89E-04 |
| | 1500 | 1.72E-04 | 2.93E-04 | 2.62E-04 | 3.91E-04 | 2.71E-04 | 3.85E-04 |
| | 2000 | 2.53E-04 | 3.96E-04 | 3.16E-04 | 4.73E-04 | 5.32E-04 | 4.94E-04 |
| | 2500 | 6.04E-04 | 5.11E-04 | 7.03E-04 | 4.78E-04 | 9.01E-04 | 3.13E-04 |
| | 3000 | 7.03E-04 | 4.78E-04 | 5.05E-04 | 5.22E-04 | 1.00E-03 | 0.00E+00 |
| TopFP | 500 | 5.50E-03 | 4.74E-03 | 3.70E-03 | 4.35E-03 | 9.10E-03 | 2.85E-03 |
| | 1000 | 8.20E-03 | 3.79E-03 | 1.90E-03 | 2.85E-03 | 7.30E-03 | 4.35E-03 |
| | 1500 | 8.20E-03 | 3.79E-03 | 3.70E-03 | 4.35E-03 | 9.10E-03 | 2.85E-03 |
| | 2000 | 1.00E-02 | 0.00E+00 | 1.00E-03 | 0.00E+00 | 9.10E-03 | 2.85E-03 |
| | 2500 | 7.30E-03 | 4.35E-03 | 1.00E-03 | 0.00E+00 | 8.20E-03 | 3.79E-03 |
| | 3000 | 7.30E-03 | 4.35E-03 | 1.00E-03 | 0.00E+00 | 9.10E-03 | 2.85E-03 |
| MACCS | 500 | 1.00E-02 | 0.00E+00 | 3.10E-02 | 4.65E-02 | 1.00E-02 | 0.00E+00 |
| | 1000 | 1.00E-02 | 0.00E+00 | 9.10E-02 | 2.83E-02 | 1.00E-02 | 0.00E+00 |
| | 1500 | 1.00E-02 | 0.00E+00 | 1.00E-01 | 0.00E+00 | 1.00E-02 | 0.00E+00 |
| | 2000 | 8.20E-03 | 2.84E-03 | 1.00E-01 | 0.00E+00 | 5.50E-02 | 4.47E-02 |
| | 2500 | 2.80E-03 | 3.74E-03 | 1.00E-01 | 0.00E+00 | 9.01E-02 | 3.11E-02 |
| | 3000 | 1.00E-03 | 0.00E+00 | 1.00E-01 | 0.00E+00 | 9.01E-02 | 3.11E-02 |
| Morgan | 500 | 4.00E-04 | 4.72E-04 | 7.10E-03 | 4.57E-03 | 5.10E-03 | 4.86E-03 |
| | 1000 | 3.00E-03 | 4.72E-03 | 1.00E-02 | 0.00E+00 | 9.10E-03 | 2.83E-03 |
| | 1500 | 1.00E-02 | 0.00E+00 | 1.90E-02 | 2.83E-02 | 1.00E-02 | 0.00E+00 |
| | 2000 | 1.00E-02 | 0.00E+00 | 1.00E-02 | 0.00E+00 | 1.00E-02 | 0.00E+00 |
| | 2500 | 1.00E-02 | 0.00E+00 | 1.00E-02 | 0.00E+00 | 1.00E-02 | 0.00E+00 |
| | 3000 | 1.00E-02 | 0.00E+00 | 1.00E-02 | 0.00E+00 | 1.00E-02 | 0.00E+00 |

**Table B2.** Values of the optimal KRR hyperparameter $\gamma$ obtained by cross-validation as a function of descriptor type and training set size. The procedure was repeated 10 times with re-shuffled data. Average values ($\bar{\gamma}$) were used in further KRR models. We also report the statistical standard deviation $\Delta\gamma$.

| Descriptor | Training Set Size | $K_{WIOM/G}$ | | $K_{W/G}$ | | $P_{sat}$ | |
|---|---|---|---|---|---|---|---|
| | | $\bar{\gamma}$ | $\Delta\gamma$ | $\bar{\gamma}$ | $\Delta\gamma$ | $\bar{\gamma}$ | $\Delta\gamma$ |
| CM | 500 | 1.00E-04 | 0.00E+00 | 1.00E-04 | 0.00E+00 | 1.00E-04 | 0.00E+00 |
| | 1000 | 1.00E-04 | 0.00E+00 | 1.00E-04 | 0.00E+00 | 1.00E-04 | 0.00E+00 |
| | 1500 | 1.00E-04 | 0.00E+00 | 1.00E-04 | 0.00E+00 | 1.00E-04 | 0.00E+00 |
| | 2000 | 1.00E-04 | 0.00E+00 | 1.00E-04 | 0.00E+00 | 1.00E-04 | 0.00E+00 |
| | 2500 | 1.00E-04 | 0.00E+00 | 1.00E-04 | 0.00E+00 | 1.00E-04 | 0.00E+00 |
| | 3000 | 1.00E-04 | 0.00E+00 | 1.00E-04 | 0.00E+00 | 1.00E-04 | 0.00E+00 |
| MBTR | 500 | 5.30E-05 | 5.30E-05 | 7.30E-05 | 4.35E-05 | 5.40E-05 | 4.86E-05 |
| | 1000 | 8.20E-05 | 8.20E-05 | 1.00E-04 | 0.00E+00 | 1.81E-04 | 2.89E-04 |
| | 1500 | 1.90E-04 | 1.90E-04 | 2.80E-04 | 3.79E-04 | 2.80E-04 | 3.79E-04 |
| | 2000 | 2.80E-04 | 2.80E-04 | 3.70E-04 | 4.35E-04 | 5.50E-04 | 4.74E-04 |
| | 2500 | 6.40E-04 | 6.40E-04 | 7.30E-04 | 4.35E-04 | 9.10E-04 | 2.85E-04 |
| | 3000 | 7.30E-04 | 7.30E-04 | 5.50E-04 | 4.74E-04 | 1.00E-03 | 0.00E+00 |
| TopFP | 500 | 1.00E-04 | 0.00E+00 | 9.10E-05 | 2.85E-05 | 1.00E-04 | 0.00E+00 |
| | 1000 | 1.00E-04 | 0.00E+00 | 1.00E-04 | 0.00E+00 | 1.00E-04 | 0.00E+00 |
| | 1500 | 1.00E-04 | 0.00E+00 | 1.00E-04 | 0.00E+00 | 1.00E-04 | 0.00E+00 |
| | 2000 | 1.00E-04 | 0.00E+00 | 1.00E-04 | 0.00E+00 | 1.00E-04 | 0.00E+00 |
| | 2500 | 1.00E-04 | 0.00E+00 | 1.00E-04 | 0.00E+00 | 1.00E-04 | 0.00E+00 |
| | 3000 | 1.00E-04 | 0.00E+00 | 1.00E-04 | 0.00E+00 | 1.00E-04 | 0.00E+00 |
| MACCS | 500 | 1.00E-02 | 0.00E+00 | 9.10E-02 | 2.83E-02 | 1.00E-02 | 0.00E+00 |
| | 1000 | 1.00E-02 | 0.00E+00 | 9.10E-02 | 2.83E-02 | 1.00E-02 | 0.00E+00 |
| | 1500 | 1.00E-02 | 0.00E+00 | 1.00E-01 | 0.00E+00 | 1.00E-02 | 0.00E+00 |
| | 2000 | 1.00E-02 | 0.00E+00 | 1.00E-01 | 0.00E+00 | 5.50E-02 | 4.47E-02 |
| | 2500 | 1.00E-02 | 0.00E+00 | 1.00E-01 | 0.00E+00 | 9.10E-02 | 2.83E-02 |
| | 3000 | 1.00E-02 | 0.00E+00 | 1.00E-01 | 0.00E+00 | 9.10E-02 | 2.83E-02 |
| Morgan | 500 | 4.00E-04 | 4.72E-04 | 9.00E-03 | 3.14E-03 | 5.10E-03 | 4.86E-03 |
| | 1000 | 3.00E-03 | 4.72E-03 | 1.00E-02 | 0.00E+00 | 9.01E-03 | 3.11E-03 |
| | 1500 | 1.00E-02 | 0.00E+00 | 1.00E-02 | 0.00E+00 | 1.00E-02 | 0.00E+00 |
| | 2000 | 1.00E-02 | 0.00E+00 | 1.00E-02 | 0.00E+00 | 1.00E-02 | 0.00E+00 |
| | 2500 | 1.00E-02 | 0.00E+00 | 1.00E-02 | 0.00E+00 | 1.00E-02 | 0.00E+00 |
| | 3000 | 1.00E-02 | 0.00E+00 | 1.00E-02 | 0.00E+00 | 1.00E-02 | 0.00E+00 |

The parameter $s_k$ effectively tunes the cutoff distance. The functions $\mathrm{MBTR}_k(x)$ are then discretized with $n_k$ many points in the respective intervals $[x_{\min}^k, x_{\max}^k]$.

*Author contributions.* EL performed all computational work. ML advised the computations. PR, HV and TK conceived the study. All authors participated in drafting the manuscript.

*Competing interests.* The authors declare that they have no conflict of interest.

*Acknowledgements.* This work was supported by the Academy of Finland (Project numbers 315600 and 316601) and through their Flagship programme: Finnish Center for Artificial Intelligence FCAI. This work was further supported by the European Research Council project 692891-DAMOCLES, by COST (European Cooperation in Science and Technology) Action 18234 and by the University of Helsinki Faculty of Science ATMATH project. We thank CSC, the Finnish IT Center for Science and Aalto Science IT for computational resources.

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
