# Peer review of "Predicting Gas-Particle Partitioning Coefficients of Atmospheric Molecules with Machine Learning"

_Atmospheric Chemistry and Physics, 2020_

## Author Comment (AC1)

**Reviewer 1**

**Reviewer's comment No. 1** — [. . .] We therefore argued that "the expertise and time required to perform quantum-chemical calculations for atmospherically relevant molecules should constitute but a minor impediment to a wider adoption" (Wang et al., 2017). I am therefore very pleased to see that with their work, Lumiaro et al. have now obliterated even this minor impediment. While it would have been possible to make COSMOtherm-based predictions for datasets much larger than the 3414 molecules in Wang et al. (2017) using "brute force" and high-performance computing resources, Lumiaro et al. demonstrate convincingly that this can be achieved with much less computational effort using machine learning approaches.

The paper is very well written and, apart from some parts of the Methods section, easily accessible to those who are not familiar with computational chemistry and machine learning approaches.

**Authors' reply**:    We thank the reviewer for their positive assessment of our work!

**Reviewer's comment No. 2** — The compounds to which the trained algorithm was applied have very limited structural diversity (only normal decanes functionalized with up to six functional groups of only three types). Why was this relatively simple dataset of molecules generated, instead of using existing molecular datasets of atmospherically relevant species? For example, Valorso et al. (2011) generated > 200,000 oxidation products of a-pinene, i.e. one of the monoterpenes judged to be among "the most interesting molecules from a SOA-forming point of view" (line 307). A recent study generated datasets of 200,000, 550,000 and 750,000 atmospheric oxidation products of decane, toluene and a-pinene (Isaacman-VanWertz and Aumont, 2020).

**Authors' reply**:    At the time of our study, we were not aware of the existence, or the public availability, of the datasets suggested by the reviewer. The purpose of our admittedly simple C10 dataset was not to comprehensively evaluate the performance of the algorithm (as that would in any case required extensive further COSMOtherm calculations), but just to perform a relatively simple "sanity check" of its predictions. We completely agree with the reviewer that the actual structures of the molecules in our C10 set may not be atmospherically relevant, although functional group composition certainly is.

We have now looked into the alpha-pinene dataset suggested here, but discovered that some alpha-pinene oxidation products are already included in Wang et al's dataset for which we trained our machine learning model. Testing model predictions on the same molecule class it is trained on is not good practice in ML model validation, so we did not extend our "sanity check" to these molecules. We are now building a larger dataset with an active machine learning technique and additional COSMOtherm calculations. The new dataset is based on compounds generated with the GECKO algorithm. It will be substantially larger and atmospherically more relevant than the C10 dataset. We hope to be able to report preliminary result on this work soon in a separate publication.

We clarified our motivation behind the choice of the validation dataset in the manuscript:

"While the functional group composition of our C10 dataset is atmospherically relevant, the particular molecules are not. The purpose of this dataset is to perform a relatively simple sanity check

on the machine learning predictions, on a set of compounds structurally different from those in the training dataset. We note that using e.g. more atmospherically relevant compounds such as alpha-pinene oxidation products for this purpose might be counter-productive, since Wang *et al.*'s dataset used for training contains several such compounds."

**Reviewer's comment No. 3** — Can the authors explain in more detail how a machine-learning model that is not fed with information on the conformations of a molecule is "capable of accounting for hydrogen-bonding interactions between functional groups" (line 366). Is this merely by structural similarity with molecules within the training set that also have such capabilities?

**Authors' reply**: We agree that this must be due to structural similarity in the training set. The linear structures we generate in our work do of course not have hydrogen bonds. The hydrogen bonds could therefore only be introduced by conformers. The SMILES string for all conformers of a molecule is of course the same. So if there is something in a SMILES string that indicates to the machine learning method that the structure prefers a conformer with hydrogen bonding and representative structures are in the training set, this could indeed be learned. We have clarified this in the manuscript:

"As we did not include conformational information of our C10 molecules in the machine-learning predictions, this is most likely due to structural similarities between the C10 compounds, and hydrogen-bonding molecules in the training dataset."

**Reviewer's comment No. 4** — In this context, it is stated on line 380: "MBTR encoding requires knowledge of the 3-dimensional molecular structure, which raises the issue of conformer search", but section 2.2.2. does not spell out how that issue was resolved in the current study?

**Authors' reply**: To compute the MBTR and CM descriptors, we employed the *openbabel* software to convert the SMILES strings provided in the Wang *et al.* dataset into 3-dimensional molecular structures. Wang and collaborators must have themselves carried out a conformer search with COSMOconf, since the COSMOtherm calculations they performed typically average over many (up to 100) located conformers, but did not publish this data. Since values of KW/G, KWIOM/G and PSat were computed by averaging over conformers, there is no single conformer that correlates strongly with these values, so we decided to forgo the computationally costly conformer searches. We have now clarified this point in the manuscript:

"To compute the MBTR and CM descriptors we employed the *openbabel* software to convert the SMILES strings provided in the Wang *et al.* dataset into 3-dimensional molecular structures. We did not perform any conformer search."

**Reviewer's comment No. 5** — Can the author propose how in the future, the atmospheric community will be able to obtain predictions for atmospherically relevant molecules, i.e. how a trained machine learning algorithm or its predictions could be made available for use by others. The authors still intend to improve this algorithm by extending the "training set to encompass especially atmospheric autoxidation products" (line 388), i.e. may not yet want to make the existing version accessible to others. However, it may be instructive to hear how this could look

like eventually. Is it conceivable to create an easy-to-use software or webpage that is fed batches of SMILES and generates KW/G, KWIOM/G and PSat as calculated by the algorithm? Or would that take the form of a searchable database that has such algorithm-generated values stored for the "104 - 107 different organic compounds" (line 60) of atmospheric interest?

**Authors' reply**:    Our "role model" here is the excellent and user-friendly UManSysProp webpage, where a user can insert e.g. a SMILES string, and obtain (among other things) saturation vapor pressure predictions computed using a variety of group contribution methods. We anticipate that the user interface of our model will eventually be similar to that. Ideally, in addition to providing predicted values for the different parameters, the results would also include an estimate of how reliable the predictions are (based on how similar or different the user-input molecule is to those included in the training dataset).

**Reviewer's comment No. 6**  —  Many atmospheric applications require knowledge of phase partitioning at variable temperatures. COSMOtherm can also calculate the enthalpy of vaporization and the internal energies of phase transfer between the gas phase and water or WIOM. It would probably be advisable to eventually also train a machine learning algorithm to predict those thermodynamic properties.

**Authors' reply**:    We agree completely. We also note, related to issues raised by the other reviewers, that predictions of various activity coefficients computable by COSMOtherm could also be useful. We changed the manuscript accordingly:

"We also intend to extend the machine learning model to predict a larger set of parameters computed by COSMOtherm, such as vaporization enthalpies, internal energies of phase transfer, and acivity coefficients in representative phases."

**Reviewer's comment No. 7**  —  I find Figure 2 not particularly useful. While it could be beneficial to have a representation of the machine learning workflow, it should look less generic than what is depicted here. For example, "representations" make no appearance in that diagram, but are obviously an important part of the process. Also, the training and testing of the machine learning algorithm is presumably a key element of the workflow.

**Authors' reply**:    We changed the figure following the referee's recommendation.

[Figure]

Figure 1: Schematic of our machine learning workflow: The raw input data is converted into molecular representations (referred to as features in this figure). We then set up and train a machine learning method. After evaluating its performance in step 5, we may adjust the features. Once the machine learning model is calibrated and trained, we make predictions on new data.

**Reviewer's comment No. 8** — Footnote on page 2: While it is indeed quite common to estimate the KO/G by dividing KO/W by KG/W (e.g. Meylan and Howard, 2005) this is only an approximation. Whereas the octanol phase in a KO/W measurement is saturated with water and the aqueous phase is saturated with octanol, the solvents in a KW/G and KO/G measurement are typically pure. This can lead to a failure of the thermodynamic triangle to correctly estimate KO/G for hydrophobic substances (Beyer et al. 2002).

**Authors' reply**: Thank you for the clarification! We have changed the footnote to: "The gas-octanol partitioning coefficient ($K_{O/G}$) can then to good approximation be obtained from these by division."

**Reviewer's comment No. 9** — Line 96. The abbreviation KRR is used here for the first time, but is only introduced on line 106.

**Authors' reply**: We removed the first instance of KRR, since it was not required on line 96.

**Reviewer's comment No. 10** — Line 134: bromine not bromide

**Authors' reply**: Fixed

**Reviewer's comment No. 11** — Line 146: The Pyzer-Knapp et al. reference is missing the year "2015" (also in the reference list)

**Authors' reply**: Added

**Reviewer's comment No. 12** — Line 154: What does it mean if a molecular representation is "continuous"?

**Authors' reply**: A molecular representation is continuous, if continuous changes in the molecular structure translate into continuous changes in the representation. The many-body tensor representation (MBTR) is a good example for a continuous representation, whereas the Coulomb matrix (CM) is discontinuous. Both encode inverse distances. The MBTR does so by Gaussian broadening each inverse distance between atom pairs and then summing up these Gaussians in separate vectors for each atomic species pair. Small changes in the interatomic distances then lead to small changes in the Gaussian peak positions. Conversely, the CM assigns one value to each atom pair and collects those in a matrix whose rows and columns are sorted by their respective norm. A small interatomic distance variation could then lead to an exchange of rows and columns, which is not a continuous change of the representation.

**Reviewer's comment No. 13** — Line 320: Explain the meaning of "cheaper to evaluate".

**Authors' reply**: The MBTR descriptor has a large data structure (22,400 vector elements) and was evaluated in several calculation stages. In contrast, TopFP is represented by a smaller data

structure (8,192 vector elements) and required less computational time to evaluate, also because it did not need the conversion to 3-dimensional structures. We have now clarified in the manuscript that by "cheaper" we refer to computational resources involved.

**Reviewer's comment No. 14** — Line 331-332: I find this sentence very confusing and I wonder whether "or less" at the end of line 331 should be deleted.

**Authors' reply**: The second "or less" was a typo, which we removed in the revised version. Thank you for spotting it!

**Reviewer's comment No. 15** — Line 336: "by almost a factor of 4000".

**Authors' reply**: "a" added as suggested

**Reviewer's comment No. 16** — Line 397 and 398: If "Zenodo, 2020" and "Gitlab, 2020" are references, they are missing from the reference list. Wouldn't it be better to provide complete links to those datasets?

**Authors' reply**: We have now updated these citations with full reference links, and DOIs where appropriate.

---

## Author Comment (AC2)

**Reviewer 2**

**Reviewer's comment No. 1** — It was interesting to read this manuscript. The topic of the manuscript is the prediction of saturation vapor pressures and partitioning coefficients between the gas phase and an aqueous phase and an organic phase respectively relevant in atmospheric science. There is a lack of experimental data on such properties and given the overwhelming amount of different molecules in the atmosphere, reliable computational methods that can predict such properties for a large amount of molecules are valuable. In this work, the authors explore the use of a machine learning method to predict selected thermodynamic properties for a large number of molecules, which seems very promising and timely.

**Authors' reply**: We thank the reviewer for their interest in our work and their constructive feedback!

**Reviewer's comment No. 2** — References: I do not find that there are enough references to the literature throughout the introduction. As an example statements like "They scatter and absorb solar radiation and form cloud droplets in the atmosphere, affect visibility and human health and are responsible for large uncertainties in the study of climate change." and "Most aerosol particles are secondary organic aerosols" should be accompanied by one or more literature references. Likewise, in section 4 on prediction I miss examples and references for the statements for example on functionalization and fragmentation.

**Authors' reply**: We added more literature references to the revised manuscript as suggested:

They scatter and absorb solar radiation and form cloud droplets in the atmosphere, affect visibility and human health and are responsible for large uncertainties in the study of climate change (IPCC 2013).

Most aerosol particles are secondary organic aerosols (SOAs) that are formed by oxidation of volatile organic compounds (VOCs), which are in turn emitted into the atmosphere for example from plants or traffic (Shrivastava et al. 2017).

Many of the most interesting molecules from a SOA-forming point of view, e.g. monoterpenes, have around 10 carbon (Zhang et al. 2018).

Atmospheric oxidation reaction mechanisms can be generally classified into two main types: fragmentation and functionalization (Kroll et al. 2009, Seinfeld et al. 2016).

With the following references:

IPCC 2013: IPCC, 2013: Climate Change 2013: The Physical Science Basis. Contribution of Working Group I to the Fifth Assessment Report of the Intergovernmental Panel on Climate Change, Stocker, T.F., D. Qin, G.-K. Plattner, M. Tignor, S.K. Allen, J. Boschung, A. Nauels, Y. Xia, V. Bex and P.M. Midgley (eds.), Cambridge University Press, Cambridge, United Kingdom and New York, NY, USA, 1535 pp.

Kroll et al. 2009: Kroll, J. H., Smith, J. D., Che, D. L., Kessler, S. H., Worsnop, D. R., and Wilson, K. R.: Measurement of fragmentation and functionalizationpathways in the heterogeneous oxidation of oxidized organic aerosol, Phys. Chem. Chem. Phys., 11, 8005–8014, 2009.

Shrivastava et al. 2017: Recent advances in understanding secondary organic aerosol: Implications for global climate forcing, Rev. Geophys., 55, 509–559

Seinfeld et al. 2016: Seinfeld, J. H. and Pandis, S. N.: Atmospheric Chemistry and Physics: From Air Pollution to Climate Change, 3rd Edition, Wiley, 2016.

Zhang et al. 2018: Monoterpenes are the largest source of summertime organic aerosol in the southeastern United States, Proc. Natl. Acad. Sci. U.S.A., 115, 2038–2043, 2018

**Reviewer's comment No. 3** — The thermodynamic basis – vapor pressures and partitioning coefficients: I expect several of the low volatile species will be solids at room temperature and likely exist in the subcooled liquid state in the atmosphere. There can be a large difference between the vapor pressure of the solid and that of the subcooled liquid. I assume the vapor pressures calculated are for the subcooled liquid state. This should be specified. Likewise, it should be better explained to the reader what the physical meaning of the partitioning coefficients is? Do they represent partitioning over a flat surface? It says they are infinite dilutions – does this mean the activity coefficients are one? What values are assumed for the activity coefficients? partitioning in the atmosphere depends on many things including particle size, amount of condensed material, accommodation coefficients – I suggest this is recognized and addressed.

**Authors' reply**:   The vapor pressures are computed for the subcooled liquid state, and the partitioning coefficients correspond to flat surfaces. This has been clarified in the manuscript. Concerning these and several further issues raised by the reviewer related to the thermodynamic parameters discussed here, we would like to point out that no actual calculations on saturation vapor pressures, partitioning coefficients, etc were performed in this study. We have simply used machine learning tools to teach an algorithm to predict these parameters. All the actual thermodynamic data used in our study were taken directly from the Wang et al paper.

While the origin, quality and features of the data are of course all relevant issues, the purpose of our manuscript is to test which (if any) combinations of molecular descriptors and machine learning algorithms can be used to construct a sufficiently accurate and robust predictive model. This selection and validation of descriptors and algorithms is by no means a trivial task. While we aim to provide the reader with a general description of the underlying data, rather than just referring to Wang et al 2017 for all details, we believe that detailed derivations of each equation, or an extensive description of the exact details of all stages of a COSMOtherm calculation, are beyond the scope of this paper. Having said that, we would like to clarify that the definition of "partitioning coefficients" used here (or, to be more precise, in the COSMOtherm program as well as in the study of Wang et al) corresponds more to that used in conventional organic chemistry (for equilibrium partitioning of a solute between two bulk phases in contact with each other) than that used in atmospheric chemistry and physics. The reviewer is of course completely correct that predicting actual partitioning between a real aerosol particle and the gas phase requires the estimation of many additional thermodynamic as well as kinetic parameters, which are not considered here. A note on this has been added to the manuscript.

"For technical details on the COSMOtherm calculations performed by Wang et al., we refer to the COSMOtherm documentation (Klamt and Eckert, 2000), (Klamt, 2011), and a recent study by (Hyttinen et al.,2020), where the conventions, definitions and notations used in COSMOtherm are connected to those more commonly employed in atmospheric physical chemistry. We note especially that the saturation vapor pressures computed by COSMOtherm correspond to the subcooled liquid state, and that the partitioning coefficients correspond to partitioning between two flat bulk surfaces in contact with each other. Actual partitioning between, e.g., aerosol particles and the gas phase will depend on further thermodynamic and kinetic parameters, which are not included here."

Klamt, A. and Eckert, F.: COSMO-RS: a novel and efficient method for the a priori prediction of thermophysical data of liquids, Fluid Phase Equilib., 172, 43 – 72, 2000.

Klamt, A.: The COSMO and COSMO-RS solvation models, WIREs Comput. Mol. Sci., 1, 699–709, 2011.

Hyttinen, N., Elm, J., Malila, J., Calderón, S. M., and Prisle, N. L.: Thermodynamic properties of isoprene- and monoterpene-derived515organosulfates estimated with COSMOtherm, Atmos. Chem. Phys., 20, 5679–5696, 2020

**Reviewer's comment No. 4** — Where does the formula for calculation of saturation vapor pressure come from? Please give a derivation or a reference. The saturation vapor pressure is a property of the pure component – but here it seems to depend on the activity in a mixture and a partitioning coefficient? The equilibrium vapor pressure over a mixture depends on the activity?

**Authors' reply**: The reviewer is of course correct that the saturation vapor pressure is a property of the pure compound, and does not depend on an activity or a partitioning coefficient. The equation on line 120 is simply a way to connect partitioning coefficients (as defined by COSMOtherm, and in a certain medium, water in this example) to saturation vapor pressures. The activity coefficient is present precisely because the partitioning coefficient depends on the activity (in that medium) while the saturation vapor pressure does not. This has now been clarified in the manuscript, and we have also rearranged the equation so that it is solved for the partitioning coefficient instead (thus illustrating that the partitioning coefficient depends on the saturation vapor pressure rather than vice versa). The exact details of how saturation vapor pressures are calculated by COSMOtherm are fairly complicated, and - as mentioned above - beyond the scope of this manuscript given that all the actual thermodynamic data are taken directly from Wang *et al*. However, we have added references to both the COSMOtherm documentation, and to a recent study by Hyttinen et al, where the COSMOtherm approach for calculating various thermodynamic parameters is expressed using terms and definitions more familiar to atmospheric physical chemists.

"This illustrates that unlike the saturation vapor pressure $P_{\text{sat}}$, which is a pure-compound property, the partitioning coefficient also depends on the activity of the molecule in the chosen liquid solvent, in this case water."

"See (Hyttinen et al., 2020) for a discussion on the connection between different conventions and the notation used by COSMOTherm, and those commonly employed in atmospheric physical chemistry."

**Reviewer's comment No. 5** — What is meant with the statement "Saturation vapor pressure

describes the interaction of a compound with itself" (page 2 line 29/30) ? and "partitioning coefficients (K) for the interaction of the compound with representative other species." I would say, that it is the activity coefficients that account for interactions between molecules in the condensed phase. In the gas phase – do the authors consider molecular interactions?

**Authors' reply**: Our formulation, especially the use of the verb "describes", may have been poor – we were simply trying to convey exactly what the reviewer stated in the previous comment, i.e. that the saturation vapor pressure is a pure-compound property, and depends only on how a compound interacts with itself (i.e. NOT on how it interacts with any other compounds). We agree that interactions with other compounds is described (or accounted for) by activity coefficients. In the conceptual framework used here, as illustrated for example by the equation on line 120 discussed above (with the added caveat that the saturation vapor pressure is indeed a pure-compound property), the partitioning coefficients depend on the activity coefficients. We have reformulated the text and added explicit mention of this to the manuscript. COSMOtherm does not consider intermolecular interactions in the gas phase. This is justified as the mean free path in atmospheric conditions is quite large. Intramolecular interactions such as H-bonds are accounted for (albeit sometimes inaccurately).

"These include the (liquid or solid) saturation vapour pressure, and various partitioning coefficients (K) in representative solvents such as water or octanol. The saturation vapor pressure is a pure-compound property, which essentially describes how efficiently a molecule interacts with other molecules of the same type. In contrast, partitioning coefficients depend on activity coefficients, which encompass the interaction of the compound with representative solvents."

**Reviewer's comment No. 6** — Some sentences are unclear: eg. "For relatively simple organic compounds, efficient empirical parametrizations have been developed to predict their condensation-relevant properties. " – the authors should help the reader here with more clear definitions - what is a "relatively simple organic compound" – and what are the exact condensation relevant properties and which efficient empirical parameterizations are the authors referring to here (references should be given) ?

**Authors' reply**: By relatively simple we mean relatively few functional groups, typically four or less. However, this quantification depends somewhat on the compound families, e.g. for peroxides the parametrisation datasets of the currently available approaches rarely contain data for compounds with even two functional groups. This has now been clarified. By condensation-relevant properties we here mean primarily saturation vapor pressures, as well as partitioning coefficients. This has also been clarified. The parametrizations we are referring to are listed in the next sentences (starting with "These include"). We give here in total 8 references to empirical parametrizations, plus one reference to a user-friendly interface. The connection between the beginning and end of the paragraph in question has been clarified by changing "These" to "Such parametrizations".

"For relatively simple organic compounds, typically with up to three or four functional groups, efficient empirical parametrizations have been developed to predict their condensation-relevant properties, for example saturation vapor pressures. Such parameterizations include..."

**Reviewer's comment No. 7** — To help the reader I also suggest to restructure the manuscript a bit and define the coefficients that are modelled already in the introduction.

**Authors' reply**:   The relevant coefficients are already defined in the first paragraph of the introduction:

"Typical partitioning coefficients in chemistry include ($K_{W/G}$) for the partitioning between the gas phase and pure water (i.e. an infinitely dilute solution of the compound), and ($K_{O/W}$) for the partitioning between octanol and water solutions. For organic aerosols, the partitioning coefficient between the gas phase and a model water-insoluble organic matter phase (WIOM; $K_{WIOM/G}$) is more appropriate than ($K_{O/G}$)."

**Reviewer's comment No. 8**  —  How was vapor pressures obtained/calculated from COSMOtherm – this is unclear from the manuscript and should be specified.

**Authors' reply**:   As described in response to previous questions, we added references to both the COSMOtherm documentation, which explains in detail how the vapor pressures are obtained, and to Hyttinen *et al.* who connect the COSMOtherm approach to concepts and definitions more familiar to atmospheric physical chemists. Since we have not performed any actual COSMOtherm calculations in this work, and since the derivations in question are multiple pages long (each), we have not reproduced them in this manuscript. On this topic, we refer the reviewer to the Wang et al. (2017) manuscript.

**Reviewer's comment No. 9**  —  Could the authors reflect on why the MBTR method performs so much better than the other methods?

**Authors' reply**:   We do address this point in the conclusion section of the manuscript:

"KRR is a relatively simple kernel-based machine-learning technique that is straightforward to implement and fast to train. Given model simplicity, the quality of learning depends strongly on information content of the molecular descriptor. More specifically, it hinges on how well each format encapsulates the structural features relevant to the atmospheric behaviour. The exhaustive approach of MBTR descriptor to documenting molecular features has led to very good predictive accuracy inmachine learning of molecular properties (Stuke et al., 2019; Langer et al., 2020; Rossi and Cumby, 2020; Himanen et al., 2020) and this work is no exception. The lightweight CM descriptor does not perform nearly as well, but these two representations from physical sciences provide us with an upper and lower limit on predictive accuracy."

In short, the MBTR is a much larger descriptor than the Coulomb matrix or the ChemInformatics fingerprints. It not only captures the topology of an organic molecule, like the fingerprints, but also includes the additional information provided by inter-atomic distances and bond angles. Generally speaking, the more relevant information is encoded in the descriptor, the better the machine learning.

**Reviewer's comment No. 10**  —  Accuracy and performance: It should be stated explicitly what the COSMOtherm accuracy is, both on the predicted saturation vapor pressures and on the partitioning coefficients.

**Authors' reply**:   First, we note again that the purpose of our study was to test which combinations of molecular descriptors and machine learning algorithms produce accurate predictive

models for (e.g.) saturation vapor pressures of polyfunctional molecules. We only used COSMOtherm data because of the limited availability of relevant experimental data. The accuracy of the COSMOtherm data itself, while not irrelevant, is not particularly crucial for this study.

Having said this, we certainly agree that it would be extremely desirable to know the COSMOtherm accuracy for a given polyfunctional molecule. Sadly, reliably estimating this accuracy is extremely challenging, primarily due to the lack of measured saturation vapor pressures for extremely low-volatility polyfunctional compounds, as also mentioned by the reviewer in the first paragraph of their comment. Lack of experimental data, on the other hand, is one of the main reasons why COSMOtherm calculations are useful. We note that this is a general problem with applied quantum chemistry: the methods are scientifically the most useful for computing values which cannot (yet) be measured, but this same lack of measurements precludes an accurate assessment of error margins for the actual calculation of interest.

The COSMOtherm documentation and literature give some accuracy guidelines, for example Eckert and Klamt (2002; see manuscript for reference) report that the maximum deviation for the saturation vapor pressure predicted for the 310 compounds included in the original COSMOtherm parametrization dataset is a factor of 3.7. In principle, the parameters of COSMOtherm should be element-specific, not compound-specific, but in practice this does not really hold for the H-bonding parameters, as alluded to also by reviewer number 3. Our own calculations for complex atmospherically relevant polyfunctional molecules (see e.g. Kurtén et al., 2018) indicate that the error margins are likely to be considerably larger than this factor of 3.7. For complex polyfunctional molecules, especially ones capable of forming intra-molecular hydrogen bonds, we further find that the accuracy of the values depend on the details of the conformational sampling. As a very rough estimate, based on direct comparisons to the very limited number of available experiments on relevant compounds (Kurtén et al 2018, Krieger et al 2018), the error margin of the computed saturation vapor pressures are probably around an order of magnitude for moderately complex (2-3 functional groups) molecules, possibly increasing by as much as a factor of 5 per each potential intra-molecular hydrogen bond. A similar error margin was used in very a recent study by Hyttinen et al (J. Phys. Chem. A 2021, in press, https://doi.org/10.1021/acs.jpca.0c11328). The error margins of the partitioning coefficients are likely somewhat smaller, as argued by Wania et al (2014). This has now been noted in the manuscript as requested.

"While the maximum deviation for the saturation vapor pressure predicted for the 310 compounds included in the original COSMOtherm parametrization dataset is only a factor of 3.7 (Eckert and Klamt, 2000), the error margins increase rapidly especially with the number of intramolecular hydrogen bonds. In a very recent study, Hyttinen *et al.* estimated that the uncertainty of the COSMOtherm saturation vapor pressure and partitioning coefficient predictions increases by a factor of 5 for each additional intra-molecular hydrogen bond (Hyttinen 2021)."

Hyttinen, N., Wolf, M., Rissanen, M. P., Ehn, M., Peräkylä, O., Kurtén, T., and Prisle, N. L.: Gas-to-Particle Partitioning of Cyclohexene-and $\alpha$-Pinene-Derived Highly Oxygenated Dimers Evaluated Using COSMOtherm, J. Phys. Chem. A (2021), in press.

**Reviewer's comment No. 11** — Page 7 line 158 – what is "good performance" ?

**Authors' reply**: We removed the statement, since it was not necessary in the "representation section".

**Reviewer's comment No. 12** — I miss a short description of which parent VOCs were considered for the basis set used.

**Authors' reply**: We are not completely sure what the reviewer means with this statement. As noted above, we have not computed any new thermodynamic parameters in this study. We use data from Wang et al., who in turn used the approx. 3400 molecules included in the MCM dataset at the time of their study. The parent VOCs for the MCM dataset can be seen e.g. here (`http://mcm.leeds.ac.uk/MCM/roots.htt`, and include most of the atmospherically relevant small alkanes (methane, ethane, propane etc), alcohols, aldehydes, alkenes, ketones and aromatics, as well as chloro- and hydrochlorocabons, esters, ethers, and a few representative larger VOCs such as three monoterpenes and one sesquiterpene. Some inorganics (by definition not VOCs) are also included. A brief description of the MCM dataset is now included in the manuscript. If the reviewer is referring to our C10 dataset, used solely for a preliminary "sanity check" as discussed below and in the reply to reviewer 1, then the "parent VOC" is simply n-decane.

We revised the manuscript as follows:

"The parent VOCs for the MCM dataset include most of the atmospherically relevant small alkanes (methane, ethane, propane etc), alcohols, aldehydes, alkenes, ketones and aromatics, as well as chloro- and hydrochlorocabons, esters, ethers, and a few representative larger VOCs such as three monoterpenes and one sesquiterpene. Some inorganics are also included."

**Reviewer's comment No. 13** — Regarding the prediction section. As the authors write monoterpenes are relevant molecules and as I understand the choice of 10 carbon atoms is based on monoterpenes. The choice of a linear alkane chain is motivated by simplicity – but is it relevant in the atmosphere from monoterpene oxidation? Are all the molecules studied in the master chemical mechanism? – I would have expected at least some molecules with a ring structure included.

**Authors' reply**: Please see our reply to reviewer 1 concerning this same topic. The purpose of the C10 dataset was simply to perform a basic "sanity check" of our machine-learing set-up. We purposefully chose a rather simplistic set of structures with no direct atmospheric relevance. This very feature on the other hand means that the molecules are quite different from those included in the Wang et al dataset, making our test more robust. We are in the process of performing new COSMOtherm calculations, and associated machine learning (building on the testing and validation performed here), on a much larger, more complex, and also more atmospherically relevant dataset.

**Reviewer's comment No. 14** — The authors several times discuss formation of particles and – is there a reference for some thought of threshold vapor pressure value ? For example Page 2 line 50 a threshold value of 10-12 Pa for nucleation is given.

**Authors' reply**: The exact threshold of course depends on the conditions, including both the temperature, the formation mechanism and formation rate of the molecule in question, and the concentration of pre-existing large particles. In the typical volatility classification scheme used in atmospheric chemistry and physics (VOC - SVOC - LVOC and so on), the threshold for "effectively non-volatile" has gradually crept down over the past decades, with new categories being added: first

12

ELVOC, (with E standing for "extreme") and now ULVOC (with U standing for "Ultra"). Again, precise threshold values for these definitions also vary somewhat between sources (and are anyway usually defined in terms of saturation mass concentrations rather than vapor pressures). The 10-12 kPa value (note, kPA not Pa) quoted on page 2 represents a fairly safe threshold for participation in early growth - for actual nucleation even lower volatilities would typically be needed. This has now been clarified further in the manuscript, and a reference has been added:

"If the saturation vapour pressure of an organic compound is lower than approx. 10-12 kPa, then it could condense irreversibly onto preexisting nanometer-sized cluster (Bianchi et al., 2019)."

Bianchi, F., Kurtén, T., Riva, M., Mohr, C., Rissanen, M. P., Roldin, P., Berndt, T., Crounse, J. D., Wennberg, P. O., Mentel, T. F., Wildt, J.,Junninen, H., Jokinen, T., Kulmala, M., Worsnop, D. R., Thornton, J. A., Donahue, N., Kjaergaard, H. G., and Ehn, M.: Highly Oxygenated Organic Molecules (HOM) from Gas-Phase Autoxidation Involving Peroxy Radicals: A Key Contributor to Atmospheric Aerosol, Chem. Rev., 119, 3472–3509, 2019

**Reviewer's comment No. 15** — In the abstract it says" The resulting saturation vapor pressure and partitioning coefficient distributions were physico-chemically reasonable, and the volatility predictions for the most highly oxidized compounds were in qualitative agreement with experimentally inferred volatilities of atmospheric oxidation products with similar elemental composition."

I do not see justification for this in the manuscript. I miss examples (optimally for all the compounds) where the authors give the experimental vapor pressure, the vapor pressure obtained from a state of the art group contribution method, the COSMOtherm vapor pressure and the vapor pressure obtained using the machine learning code and discuss differences and similarities. For the lowest vapor pressures experimental data are not available. The authors should give the range of vapor pressures where the model can be compared with experimental data. It is not clear what is meant with elemental composition – normally the molecular formula or even structural formula is needed to predict a vapor pressure?

**Authors' reply**:     As noted by the reviewer, there is a great lack of experimental data on volatilites of anything but the simplest atmospherically relevant compounds. In particular, there are to our knowledge NO direct experimental measurements of the volatilities of ANY highly oxidised C10 compounds, such as the monoterpene autoxidation products referred to in our discussion. Further, as noted above, we have not performed any new COSMOtherm calculations in our paper, so COSMOtherm predictions for the C10 dataset are not available either. The three-way comparison requested by the reviewer is thus impossible. A comparison between the machine learning algorithm and the COSMOtherm predictions for the molecules calculated by Wang et al, is on the other hand very relevant, and included in the discussion.

We agree that to reliably predict a saturation vapour pressure of any particular single compound, the molecular and/or structural formula is usually needed. However, for more complex compounds such as monoterpene autoxidation products, this information is generally not available - only the elemental composition can be extracted from mass spectrometric measurements. The "inferred volatilities" discussed here are basically fits of the volatilities inferred from the measured condensation behaviour to the measured elemental compositions. While imperfect, this approach is fairly common in the literature. The point we wish to make here is that the predictions for our most

highly oxidized C10 compounds are in qualitative agreement with the predictions of such empirical fits. We have reformulated the paragraph in question to avoid giving a misleading sense of accuracy.

"The resulting saturation vapor pressure and partitioning coefficient distributions were physico-chemically reasonable, for example, in terms of the average effects of the addition of single functional groups. The volatility predictions for the most highly oxidized compounds were in qualitative agreement with experimentally inferred volatilities of, for example, alpha-pinene oxidation products with as-yet unknown structures, but similar elemental composition."

**Reviewer's comment No. 16** — Page 2 line 3: Several experimental techniques are capable of measuring saturation vapor pressures of 10-5 Pa. It would be appropriate to cite literature providing experimental vapor pressures. What is the definition of non-volatile that the authors use?

**Authors' reply**: By "non-volatile" we mean at least "ELVOC", if not "ULVOC", i.e., a molecule that does not appreciably evaporate even from a nanometer-sized particle. The threshold for this is many orders of magnitude lower than 10-5 Pa. We have added reference to a review of saturation vapor pressure measurement techniques.

"See e.g. (Bilde 2015) for a review of experimental saturation vapor pressure measurement techniques."

M. Bilde et al., Saturation Vapor Pressures and Transition Enthalpies of Low Volatility Organic Molecules of Atmospheric Relevance: From Dicarboxylic Acids to Complex Mixtures. Chem. Rev. 2015, 115, 4115-4156.

**Reviewer's comment No. 17** — Page 3 line 63: "Here, we take a different approach compared to previous parametrization studies, and consider a data-science perspective (Himanen et al., 2019). Instead of assuming chemical or physical relations, we let the data speak for itself." - what is meant with letting the data speak for itself?

**Authors' reply**: Our machine learning approach produces a data-driven model. Unlike the parameterizations that are discussed in the introduction and in a previous reviewer question, we do not use chemical or physical insight to derive an analytical expression for our model, whose few parameters are then determined by fitting. In contrast, our model has a free form (the kernel expansion). The number of expansion coefficients grows with the amount of available training data and the model changes with the data. It adapts to the training data in ways a rigid parameterization cannot.

**Reviewer's comment No. 18** — Figure 9 b: what is on the y-axis - is it a percentage? or an absolute number?

**Authors' reply**: Figure 9 b is a histogram and it shows the number of molecules that have a certain saturation vapor pressure. The y-axis is labeled correctly. We capped the y-axis at 100 to make the green and orange histograms (for molecules containing 7 or 8 O atoms) visible. As Figure

7 c shows, the total number of molecules in each bin of the C10 set is much higher (going up to ~2000). If the y-axis went up to 2000, the orange and gree subsets could not be seen.

**Reviewer's comment No. 19** — Page 16: "This result demonstrates that unlike the simplest group-contribution models (which would invariably predict that the lowest-volatility compounds in our C10 dataset should be the tetrahydroxydicarboxylic acids), both the original COSMOtherm predictions, and the machine-learning model based on them, are capable of accounting for hydrogen-bonding interactions between functional groups."

I am not sure this statement is quite fair – to my knowledge state of the art group contribution methods (e.g. those on the UMAN Sysprop webpage) include interactions – which simple group contribution methods are the authors referring to and are such simple methods being used in atmospheric simulations?

**Authors' reply**: We feel that our statement and that of the reviewer do not contradict each other. *Some* state-of-the-art group contribution methods indeed do include cross-terms for interactions. However, the *simplest* ones, such as SIMPOL, do not. We have clarified this by adding mention of SIMPOL to the sentence. SIMPOL is actually used quite extensively e.g. in studies of autoxidation products, as the very lack of cross-terms makes it more robust for very large and complex molecules, though at the expense of accuracy for compounds of medium complexity. As shown e.g. in Kurtén et al (2016), some of the more sophisticated models included in the UManSysprop website, most notably the "Nannoolal" family of approaches, may fail catastrophically when applied to certain molecules containing multiple peroxide groups. In their defence, it should be noted that the methods were never even designed to work for such compounds, and indeed some of the source literature explicitly warns against doing so. We hope that the type of approach presented and piloted in this manuscript will be able to provide the robustness of SIMPOL, combined with the greater and more molecule-specific accuracy analogous to the more sophisticated models, for a very much larger set of compounds.

"This result demonstrates that unlike the simplest group-contribution models such as SIMPOL ..."

Kurtén, T., Tiusanen, K., Roldin, P., Rissanen, M. P., Boy, M., Ehn, M. and Donahue, N. M. $\alpha$-pinene Autoxidation Products May Not Have Extremely Low Saturation Vapor Pressures Despite High O:C Ratios. Journal of Physical Chemistry A, Vol. 120, 2569-2582, 2016.

---

## Author Comment (AC3)

**Reviewer 3**

**Reviewer's comment No. 1** — The authors utilize machine learning to predict saturation vapor pressure and two equilibrium-partitioning coefficients for gas-particle partitioning. For training and validating the machine learning model they use a dataset obtained by COSMOtherm calculations of theses observables for atmospheric oxidation product molecules.

The paper is well written, the topic timely and of great interest for the readers of ACP and I recommend publishing but ask the authors to take the following comments and suggestions into account.

I have one very general concern, which does not relate to the machine learning approach presented here, but to the underlying COSMOtherm data set. The authors write (e.g. line 49 page 2) that the COSMOtherm predictions have an order of magnitude accuracy. However, for a number of compounds at low saturation vapor pressures there have been studies comparing experimental saturation vapor pressures with COSMOtherm predictions and finding much larger deviations (e.g. Bannan et al., 2017, Krieger et al. 2018). It should be pointed out that the COSMOtherm model has been "calibrated" with a parametrization dataset of known compounds, which are potentially biased to high saturation vapor pressures (Klamt et al. 1998). Therefore, the accuracy of the underlying reference data may be only several orders of magnitude for low saturation vapor pressure components.

**Authors' reply**:    We completely agree. By "order of magnitude" we meant "at best order of magnitude", to contrast with the factor of 3.7 quoted in the COSMOtherm documentation. Fortunately, proper consideration and selection of conformers, as well as improvements to the H-bonding treatment in newer versions of COSMOtherm, are slowly decreasing the disagreement between the saturation vapor pressure predictions and the limited number of experimental data points for atmospherically relevant low-volatility polyfunctionals. As noted in our reply to reviewer 2, our current best estimate, based on direct comparisons to the very limited number of available experiments on relevant compounds (see e.g. Kurtén et al 2018, Krieger et al 2018), is that the error margin of the computed saturation vapor pressures are probably around an order of magnitude for moderately complex (2-3 functional groups) molecules, possibly increasing by as much as a factor of 5 per intra-molecular hydrogen bond. This has now been noted in the manuscript as discussed above.

**Reviewer's comment No. 2** — For gas-particle partitioning, the saturation vapor pressure range from about 10-11 kPa to about 10-3 kPa is relevant (e.g. Valorso et al. 2011, or the discussion starting in the last paragraph of page 2). However, Fig. 3c shows that there are hardly any molecules in the dataset below 10-8 kPa. Actually about half of the dataset contains compounds, which will be entirely in the gas phase under atmospheric conditions. Does this pose a problem?

**Authors' reply**:    Yes, this poses a serious problem for predicting volatilities of large and complex molecules, and because of it, this study should be considered a proof-of-concept pilot for finding

appropriate combinations of descriptors and machine learning algorithms. We are in the process of performing additional COSMOtherm calculations and the corresponding machine learning on a substantially larger and much more complex set of compounds generated by the GECKO algorithm. We hope to be able to report preliminary result on this work relatively soon. Plans for future directions have been added to the conclusions - section of the manuscript.

**Reviewer's comment No. 3** — Related: the last paragraph on page 6 states that Wang's dataset is rather small for machine learning but internally consistent. I intuitively understand that this helps the machine-learning model to succeed in predicting well. However, the authors write that Sanders's dataset for 17350 Henry's law constant are not internally consistent (as Wang's dataset). But what if the Sander's data are the correct ones? What if the real world is more complex than what is predicted by COSMOtherm? Would the machine learning approaches fail because it there are no easy "rules" the machine-learning algorithm can pick out of the dataset? Would the output of a model trained with these data just produce random partitioning coefficients within the range of the data set? These questions are probably impossible to answer without doing the experiment. It would have been very interesting to see how the machine-learning model perform on the dataset of Sander, but this is clearly beyond the work presented here.

**Authors' reply**:  By Sanders's dataset "not being internally consistent" we mean primarily the fact that this (impressively large) set often contains multiple entries for the same compound (corresponding to e.g. different experimental studies, often with different methods), and the actual values can vary widely. For example for many polyols, Henry's law constants in the dataset vary by 6 orders of magnitude or more. This result can obviously not be correct, as a particular compound must have precisely one Henry's law constant at one temperature. This has been clarified in the manuscript. The other type of "internal inconsistency" (or complexity) presumably referred to by the reviewer would be e.g. strong non-additivity of the effects of various functional groups, and/or cases where very small differences in structures lead to very large differences in properties. We agree that the real world contains examples of this type of inconsistency or complexity, though typically the most extreme cases tend to be for chemical reactivity rather than physical molecular properties. Certainly such complexity also makes it more challenging to define rules for predicting properties based on structures (i.e. structure-activity or structure-property relationships). COSMOtherm is able to account for some, but probably not all, of these cases, as evidenced from the discussion on the effects of intra-molecular H-bonds also in the references cited by the reviewer. We agree that experimental methods capable of probing volatilities of very complex molecules will be needed to definitively answer the question.

On a final note, uncertainties in the data, e.g. experimental noise, can easily be taken into account in probabilistic machine learning models. We are working on such probabilistic models and will report their results in a future publication. It has to be emphasized, however, that even a noisy dataset has to be internally consistent. If Henry's law constants differ by 6 orders of magnitude for a certain compound, the dataset needs to be refined.

"For example, the Sander dataset contains several molecules with multiple entries for the same property, sometimes spanning many orders of magnitude."

**Reviewer's comment No. 4** — I find section 2.2.4 rather brief. For me – being not familiar

with the topic – it is not possible to follow despite Fig. 4d. May be extent a bit?

**Authors' reply**:    We improved the description as follows:

"TopFP first extracts all topological paths of a certain lengths. The paths start from one atom in a molecule and travel along bonds until $k$ bond lengths have been traversed as illustrated in Fig. 4d. The path depicted in the figure would be OCCO. The list of patterns produced is exhaustive: Every pattern in the molecule, up to the pathlength limit, is generated. Each pattern then serves as a seed to a pseudo-random number generator (it is "hashed"), the output of which is a set of bits (typically 4 or 5 bits per pattern). The set of bits is added (with a logical OR) to the fingerprint. The length of the bitvector, maximum and minimum possible path lengths $k_{max}$ and $k_{min}$ and the length of one hash can be optimized. "

**Reviewer's comment No. 5** — Discussion on page 16: Related to my comments above, without experimental vapor pressures for the C10 compounds being available, this discussion is interesting, but there may be surprises if experimental vapor pressures become available. I feel the authors should clearly state that the COMOtherm predictions are not validated in this pressure regime at all.

**Authors' reply**:    We agree, and this has now been stated explicitly.

"However, we caution that COSMOtherm predictions have not yet been properly validated against experiments for this pressure regime. As discussed above, we can hope for order-of-magnitude accuracy at best."

**Reviewer's comment No. 6** — Technical comment: Page 12, line 292: Figure 5 should be Fig. 3, correct?

**Authors' reply**:    Many thanks to the reviewer for catching this issue, we have now corrected it on page 12.